# Large Language Model Compression— Part 1: Weight Quantization

## Abstract

In recent years, compression of large language models (LLMs) has emerged as an important problem to enable language model deployment on resource-constrained devices, reduce computational costs, and mitigate the environmental footprint of large-scale AI infrastructure. In this paper, we lay down the foundations of LLM quantization from a convex optimization perspective and propose a quantization technique that builds on this foundation for optimum quantization outcomes. Our quantization framework, CVXQ, scales to models containing hundreds of billions of weight parameters and provides users with the flexibility to compress models to any specified model size, post-training. A reference implementation of CVXQ can be obtained from.

## 1 Introduction

Large Language Models (LLMs) have become a versatile framework for solving a large number of problems in natural language processing, from text translation and summarization to conversational AI and automatic generation of radiology reports. While LLMs have surpassed traditional methods in many of these tasks, they can involve tens or hundreds of billions of weight parameters (!), and this makes their deployment onto devices with limited resources challenging—model weights and activations far exceed the available on-chip memory so that activations need to be loaded from and saved to off-chip memory throughout inference, rendering LLM inference memory-bound (Yuan et al., 2024). This greatly hinders the usability of LLMs particularly in time-sensitive applications and exacerbates the environmental footprint of large-scale AI infrastructure required by LLMs.

One way to reduce the memory requirements of large models for inference is by compressing (that (is, simplifying) the representation of the model weights and activations after training. This can be achieved via weight pruning, quantization of activations and weights, or PCA-type dimensionality reduction of weight matrices. Out of these, quantization of weights and activation has proven to be particularly useful for compressing models to very low bit depths or arbitrary user-specified model sizes (Dettmers et al., 2022; Yao et al., 2022; Frantar et al., 2022; Frantar & Alistarh, 2022; Kim et al., 2024; Shao et al., 2024; Lee et al., 2024; Guan et al., 2024). Using state-of-the-art quantization techniques, it is now possible to compress 10–100 billion-parameter LLMs to 3–4 bits per weight on average with a negligible loss of model accuracy (Chee et al., 2024; Frantar et al., 2022; Lin et al., 2024), facilitating LLM inference on a single consumer-grade GPU for example.

Although significant advances have been made in LLM quantization recently, current approaches to model quantization still lead to considerably reduced model accuracy at low bit depths, with many methods fine-tuning model weights during quantization (Frantar et al., 2022; Lee et al., 2024; Chee et al., 2024). This makes such quantization methods less suitable for the quantization of activations during inference, where fine-tuning would lead to unacceptable delays in the inference pipeline.

Given the symmetry between weights and hidden states in matrix multiplications, achieving fast and accurate quantization of both weights and activations can be crucial for enhancing computational efficiency and prediction accuracy of LLMs, as well as for informing hardware design. This work aims to address gaps in the current model compression literature and advance compression methods further to enable accurate and efficient inference on quantized LLMs.

In this paper—the first of a three-part series—we tackle the problem of LLM compression using the

framework of convex optimization. We begin with the problem of weight quantization and analyze how a model's weights should be quantized to maximize quantized model accuracy for a given bit size. We then propose a stochastic gradient descent-type algorithm to solve this problem exactly and efficiently, post-training—in minutes for billion-parameter models and in a few hours for 10–100-billion-parameter models. Compared with the recent OPTQ family of quantization methods (Frantar et al., 2022; Frantar & Alistarh, 2022; Huang et al., 2024; Lee et al., 2024; van Baalen et al., 2024) in which weights are fine-tuned during quantization, our approach spends virtually zero time on the actual quantization once the optimum bit depths have been determined. This makes our framework also suited for quantizing intermediate activations, which can further reduce the memory footprint of batched inference. Using our bespoke mixed precision CUDA kernel (see Appendix A), we also accelerate matrix-vector multiply relative to the floating-point matrix-vector multiply of cuBLAS.

## 2 PREVIOUS WORK

Early work on neural network model quantization can be attributed to Vanhoucke et al. (2011), who demonstrated that 8-bit integer arithmetic can be used for network training and inference without incurring a significant loss of accuracy. More generally, quantization-aware training (QAT) (Zhou et al., 2017; Jacob et al., 2018; D. Zhang et al., 2018; Esser et al., 2019; Y. Choi et al., 2017; Wang et al., 2019) integrates the quantization process into training by allowing the model to adapt to the reduced precision in weights (Esser et al., 2019; Jacob et al., 2018; D. Zhang et al., 2018; Zhou et al., 2017) and activations (Y. Choi et al., 2017; Wang et al., 2019) by determining the optimum bit depth (Wang et al., 2019; D. Zhang et al., 2018) and step size (Esser et al., 2019) using back-prop to facilitate the gradient to flow through quantization operators. One shortcoming of QAT methods is that model training needs to be repeated for different quantized model sizes and accuracy, which makes them less suitable for quantizing larger models such as LLMs.

More recent quantization techniques for language and vision models aim to facilitate compression of already trained models for rapid deployment without further training (Dong et al., 2019; Chen et al., 2021; Dettmers et al., 2022; Yao et al., 2022; Frantar et al., 2022; Dettmers et al., 2023; Xiao et al., 2023; Lin et al., 2024; Kim et al., 2024; Shao et al., 2024; Lee et al., 2024). These approaches quantize model weights to 3–4 or 8 bits for integer-arithmetic-only inference (Jacob et al., 2018) using mixed bit depth quantization (Wang et al., 2019; Chen et al., 2021) or by a separate handling of outlier channels (Zhao et al., 2019) to improve the accuracy of the quantized model. Loss-aware quantization techniques (Hou & Kwok, 2018; Nahshan et al., 2020; Qu et al., 2020) seek to minimize accuracy loss in quantized models by calibrating quantization and biases on calibration data. Data-free quantization methods (Nagel et al., 2019; Xu et al., 2020; K. Choi et al., 2021; Qian et al., 2023) attempt to remove the need for real calibration data by matching the distribution of weights instead (Nagel et al., 2019) or using synthetic data in place of real calibration data (K. Choi et al., 2021).

For LLM compression in particular, an extension to the Optimum Brain Surgeon (OBS) algorithm (Hassibi & Stork, 1992) known as GPTQ (Frantar et al., 2022) was proposed for the quantization of 1–100 billion parameter models. Further recent extensions to GPTQ (Dettmers et al., 2023; Lee et al., 2024) incorporate the handling of sensitive weights by scaling or simply by retaining the original weight values similarly to (Lin et al., 2024; Xiao et al., 2023). Here, we use convex optimization for fine-granularity mixed-precision weight quantization, overcoming the combinatorial nature of determining the optimal bit depth (0, 1, …, 8 bits) per channel to better attend to channel sensitivity.

## 3 QUANTIZATION FRAMEWORK

Here, we use the task of next-token prediction in language modeling as a running example. For our purposes, the end-to-end mapping of input token embeddings to predicted next-token embeddings by a pretrained language model $f$ can be expressed in the most general form as

$$\mathbf{Z} = f(\mathbf{X}) = f(\mathbf{X}, \mathbf{\Theta}_1, \mathbf{\Theta}_2, \dots, \mathbf{\Theta}_N) = f(\mathbf{X}, \mathbf{\Theta}_1, \mathbf{\Theta}_2, \dots, \mathbf{\Theta}_N, \mathbf{B}_1, \mathbf{B}_2, \dots, \mathbf{B}_N) \qquad (1)$$

in which $\mathbf{X} \in \mathbb{R}^{L \times E}$ denotes a sequence of $L$ tokens, each of which resides in some $E$-dimensional embedding space, and $\mathbf{Z} \in \mathbb{R}^{L \times E}$, embeddings of $L$ predicted next tokens. The $m$th block of weight matrices $\mathbf{\Theta}_{mM+1}, \dots, \mathbf{\Theta}_{(m+1)M}$ and bias vectors $\mathbf{B}_{mM+1}, \dots, \mathbf{B}_{(m+1)M}$ jointly parameterize the $m$th transformer block, which refines the embeddings produced by the $(m-1)$th transformer block. In

---

**Algorithm 1.** CVXQ: Bit depth determination

1  **Input:** $f(\cdot, \Theta_1, \ldots, \Theta_N)$ (model), $\{\mathbf{X}\}$ (calibration set), R (target bit rate), $B_{max} \leftarrow 8$ (max bit depth)
2  **Output:** $B_1, \ldots, B_N$ (bit depths), $S_1, \ldots, S_N$ (weight scales) , $\mu_1, \ldots, \mu_N$ (weight means)
3  **Initialize:** $\mathbf{U} \leftarrow \text{pca\_basis}(\{\mathbf{X}\}) \in \mathbb{R}^{E \times E'}$, $\mathbf{S} \leftarrow \text{sub\_sample}(\mathbf{I}_{L \times L}) \in \mathbb{R}^{L \times L'}$, $V \leftarrow 10^{-6}$
4      $B_n \leftarrow \infty$, $G_n^2 \leftarrow 0$, $\mu_n \leftarrow \text{mean}(\Theta_n)$, $S_n \leftarrow \text{std}(\Theta_n)$, $\Theta_n^q \leftarrow \Theta_n$, $B_n^q \leftarrow B_n$, $\overline{\mathbf{X}}_n \leftarrow \mathbf{0}$ for n in $1, \ldots, N$
5  **for** iter in $1, \ldots, \text{max\_iter}$ **do**
6      **for** $\mathbf{X}$ in minibatch **do**
7          $\mathbf{Z}, \mathbf{X}_1, \ldots, \mathbf{X}_N \leftarrow f(\mathbf{X}, \Theta_1^q, \ldots, \Theta_N^q, B_1^q, \ldots, B_N^q)$
8          $\overline{\mathbf{X}}_n \leftarrow (1-\alpha)\overline{\mathbf{X}}_n + (\alpha/L)\mathbf{1}^\top \mathbf{X}_n$ **for** n in $1, \ldots, N$
9          $\Gamma_1, \ldots, \Gamma_N \leftarrow \text{autograd}(\mathbf{S}^\top \mathbf{Z} \mathbf{U}, \Theta_1^q, \ldots, \Theta_N^q)$
10         $G_n^2 \leftarrow (1-\alpha)G_n^2 + (\alpha/P_n) \text{trace}(\Gamma_n^\top \Gamma_n)$ **for** n in $1, \ldots, N$
11     **for** _ in $1, \ldots, 10$ **do**
12         $B_n \leftarrow \text{clamp}(\frac{1}{2}\log_2(G_n^2 S_n^2 / V), 0, B_{max})$ **for** n in $1, \ldots, N$
13         $V \leftarrow V + \beta(\text{sum}(P_n B_n) - (\text{sum}(P_n))R)$
14     $\Theta_n^q \leftarrow \text{compand\_quantize}(\Theta_n, B_n, S_n, \mu_n)$, $B_n^q \leftarrow B_n + (\Theta_n^q - \Theta_n)\overline{\mathbf{X}}_n$ **for** n in $1, \ldots, N$

---

practice, LLM frameworks used in language modeling also require an embedder $\Theta_0 \in \mathbb{R}^{E \times V}$ and a prediction head $\Theta_{N+1} \in \mathbb{R}^{V \times E}$ to transform between embeddings and tokens from a vocabulary of size $V$, but for now, we focus on the compression of transformer block weights as is typically done in model weight quantization work (Frantar et al., 2022; Lee et al., 2024; Lin et al., 2024).

To get a sense of the number of weight matrices and their sizes in a typical language model, the 13 billion-parameter model in the OPT family (OPT-13B) contains $N = 240$ weight matrices in blocks of $M = 6$, with each block comprising $12E^2$ weights in an embedding dimension of $E = 5120$. The embedder and prediction head are parameterized by a shared matrix containing $VE$ weights, where the vocabulary size $V = 50272$. Note that each transformer block also contains $9E$ bias parameters but due to their relative scarcity, bias parameters can be communicated losslessly and still have little to no impact on the overall compression performance (Frantar et al., 2022).

Notionally, the elements of a weight matrix $\Theta$ are continuously valued so they require quantization for efficient communication and storage. Compared with vector and lattice quantization techniques (Egiazarian et al., 2024; Gong et al., 2015; van Baalen et al., 2024), scalar quantization (Frantar et al., 2022; Lin et al., 2024) can simplify decoding and even enable operations directly on quantization indices, which obviates the need for a separate dequantization process. The mid-rise uniform scalar quantization of a weight $\theta$ at a bit-depth of $B$ bits and a step size $D$ can be expressed as

$$\theta^q(B, D) = D\big(\text{clip}\big(\text{floor}\big(D^{-1}\theta\big), -2^{B-1}, 2^{B-1} - 1\big) + 2^{-1}\big), \quad B = 0, 1, 2, \ldots \tag{2}$$

and $\theta^q(B, D) = \theta$ if $B = \infty$ (for notational convenience). The problem of compressing a model $f$ now amounts to determining the optimal bit depth $B$ and the associated quantization step size $D$ for model weights. It would be impractical, however, to determine a separate $(B, D)$ for each weight $\theta$ in the model since the cost of signaling the choice of $(B, D)$ for each one would far outweigh the bit savings derived from quantization. Typically, a single $(B, D)$ pair is used to quantize a small group of weights (an entire matrix or rows or columns thereof) in which case the cost of signaling $(B, D)$ can be borne by a group of quantized weight parameters as a negligible per-weight overhead.

## 3.1 BIT DEPTH ASSIGNMENT

Suppose we want to compress $f$ by quantizing each matrix $\Theta_n$ containing $P_n$ elements according to its own bit depth $B_n$ and step size $D_n^*(B_n)$. How should $B_n$ be decided? Roughly speaking, weights that are more sensitive to output distortion should be allotted more bits to "balance the scales" while keeping the total number of bits under a given model bit budget. We can formalize this notion by expressing the weight quantization task at hand as a constrained non-linear least-squares problem:

$$\text{minimize } d(B_1, \ldots, B_N) = \mathbb{E}_{\mathbf{X}}\big\|f(\mathbf{X}, \Theta_1^q(B_1, D_1^*(B_1)), \ldots, \Theta_N^q(B_N, D_N^*(B_N))) - f(\mathbf{X})\big\|_F^2$$

$$\text{subject to } r(B_1, \ldots, B_N) = \sum_{n=1}^N P_n B_n - \left(\sum_{n=1}^N P_n\right)R = 0 \tag{3}$$

in which $R$ denotes a user-specified average model bit depth (bit rate). This problem is reminiscent of optimal resource allocation, where the objective is to maximize some utility (or minimize output

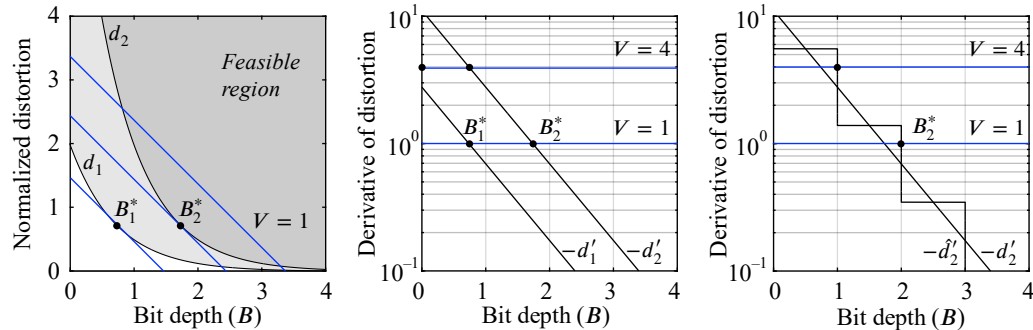

**Figure 1: Optimal bit depths.** Consider two weight matrices whose distortion functions are given by $d_1$ and $d_2$, where $d_n(B_n) = G_n^2 S_n^2 2^{-2B_n}$. For any given value of the dual variable $V$, optimal bit depths $B_1^*$ and $B_2^*$ are found where the derivative of $d_1$ and $d_2$ is $-V$, respectively (left). These points correspond to the intersections between $V$ and $-d_n' = (2 \ln 2) d_n$ (center). Integerized bit depths occur on the rounded curves $-\hat{d}_n'$ (right).

distortion in our case) by optimally spending down a given budget (the total number of bits). In this section and next, we provide insights into problem (3) and discuss its solution; see Algorithm 1.

To apply the machinery of numerical optimization to (3), we will relax the discrete constraint on the bit depths $B_1, \ldots, B_N$ while solving the problem and round the solution $B_1^*, \ldots, B_N^*$ to the nearest integers after they have been obtained. Let us write the Lagrangian of (3) as $\mathcal{L}(B_1, \ldots, B_N, V) = d(B_1, \ldots, B_N) + Vr(B_1, \ldots, B_N)$, where $V$ is a dual variable associated with the equality constraint of (3). Setting to 0 the partials of $\mathcal{L}$ with respect to $B_1, \ldots, B_N, V$ yields the optimality conditions

$$\frac{1}{P_1} \frac{\partial d(B_1, B_2, \ldots, B_N)}{\partial B_1} = \cdots = \frac{1}{P_N} \frac{\partial d(B_1, B_2, \ldots, B_N)}{\partial B_N} = -V, \qquad r(B_1, \ldots, B_N) = 0 \quad (4)$$

so, problem (3) can be solved by alternately updating the bit depths $B_1, \ldots, B_N$ (primal variables) and the trade-off $V$ (dual variable) until all optimality conditions are met. In words, the optimality conditions are reached once the marginal decrease in the output distortion from an infinitesimal bit is equal across layers at $-V$ and once we have assigned exactly $R$ bits per weight on average.

Since the quantization function (2) is constant almost everywhere, a naive computation of the partial derivatives of $d$ with respect to $B_1, \ldots, B_N$ using the chain rule of differentiation does not provide a useful direction for descent. One result from rate–distortion theory (Gersho & Gray, 1991) is that for any random variable of finite variance, quantization error decreases by half with every additional bit at a sufficiently high bit depth. More specifically to our problem, we can write (Appendix B)

$$-\frac{1}{2 \ln 2} \frac{\partial d(B_1, \ldots, B_N)}{\partial B_n} \approx \mathbb{E}_{\mathbf{X}} \left\| \frac{\partial f(\mathbf{\Theta}_1^q(B_1), \ldots, \mathbf{\Theta}_N^q(B_N))}{\partial \mathbf{\Theta}_n} \Delta_n^q(B_n) \right\|_F^2 \approx P_n H_n \underbrace{G_n^2 S_n^2 2^{-2B_n}}_{= d_n(B_n)} \quad (5)$$

in which $\mathbf{\Theta}_n^q(B_n) = \mathbf{\Theta}_n^q(B_n, D_n^*(B_n))$ for brevity, $G_n^2$ and $S_n^2$ represent the variances of the elements of $\partial_{\mathbf{\Theta}_n} f(\mathbf{X}, \mathbf{\Theta}_1^q, \ldots, \mathbf{\Theta}_N^q)$, and of $\mathbf{\Theta}_n^q$, respectively, and $H_n$ is a quantization coefficient that depends on the type of weight distribution, with $H_n = 1.42$ for Gaussian, 0.72 for Laplace, etc. (Gersho & Gray, 1991). Assuming weights are distributed similarly across layers with $H_1 = \cdots = H_N$, factors $H_n$ and constant $-\frac{1}{2 \ln 2}$ can be removed the above expression without affecting the solution of (3).

Coupled with the above closed-form expression for the partial derivatives, optimality conditions (4) naturally lend themselves to dual ascent-type methods for solving problem (3). The idea behind dual ascent (Boyd et al., 2011) is to alternately update the primal $B_1, \ldots, B_N$, and dual $V$ variables, with one set held fixed while updating the other. After initializing $B_1 = \cdots B_N = \infty$, $V$ to some small positive number, and computing $G_1^2, \ldots, G_N^2$, we update the bit depths and trade-off iteratively via

$$B_n \leftarrow \text{clamp}\left(\frac{1}{2} \log_2 \left(\frac{G_n^2 S_n^2}{V}\right), 0, B_{\max} = 8\right) \qquad \text{for } n = 1, \ldots, N$$

$$V \leftarrow V + \alpha\left(\sum_{n=1}^N P_n B_n - \left(\sum_{n=1}^N P_n\right) R\right) \tag{6}$$

in which $\alpha$ denotes a step size for dual update. Figure 1 illustrates the optimality conditions for bit

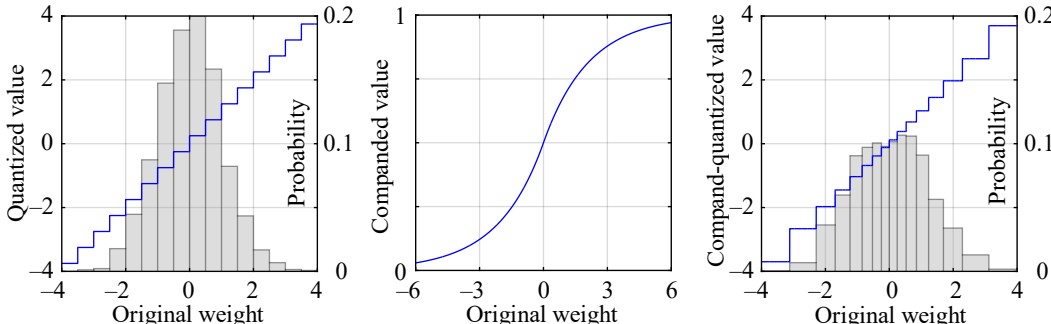

**Figure 2: Companding quantization.** Illustrated for a 4-bit quantizer (16 quantization levels) on Gaussian weights with zero mean and unit variance. Uniform quantization across the entire range of weight values (left) leads to unduly large quantization bins (hence quantization errors) for more probable weights. Companding the weights to the range $(0,1)$ prior to uniform quantization (middle) reduces quantization errors for more probable weights (right), reducing the mean square error.

depths. With $G_n^2$ and $S_n^2$ fixed, dual ascent steps (6) typically converge within a few iterations (tol = $10^{-6}$ bit, step size $\alpha = 2$) after which the obtained $B_n$ are rounded to integers. The non-linear nature of the least squares objective $d$ (3) means that iteration (6) should be repeated after the bit depths $B_n$ are updated. Using the updated $B_n$, we first obtain the re-quantized weights $\Theta_n^q(B_n)$ along with the re-computed gradient variances $G_n^2$, based on which $B_n$ can be further updated via (6).

Evaluating $\partial_{\Theta_n} f(\mathbf{X}, \Theta_1^q(B_1), \ldots, \Theta_N^q(B_N))$ across the entire calibration dataset at every iteration is prohibitively expensive given the dimensionality of the output $f(\mathbf{X}) \in \mathbb{R}^{L \times E}$ and the cost of back-propagating each element through $f$. To overcome this difficulty, we perform PCA on $f(\mathbf{X})$ along the embedding dimension (of $E$) and sub-sample along the token dimension (of $T$), and accumulate gradient variances by back-propagating only a mini-batch of calibration examples every time:

$$G_n^2 \leftarrow (1 - \beta)G_n^2 + \beta \, \mathbb{E}_{\mathbf{X} \sim \text{batch}} \left\| \frac{\partial \mathbf{S}^T f(\Theta_1^q(B_1), \ldots, \Theta_N^q(B_N))\mathbf{U}}{\partial \Theta_n} \right\|_F^2 \quad \text{for } n = 1, \ldots, N \qquad (7)$$

in which $\beta$ denotes the learning rate, and $\mathbf{S}^T$ and $\mathbf{U}$ represent the PCA projection and sub-sampling operators, respectively. In practice, we further accelerate variance accumulation by cycling through PCA coefficients and back-propagating only one coefficient per sample in every minibatch.

## 3.2 ACTUAL QUANTIZATION

Suppose now the weight matrices $\Theta_1, \ldots, \Theta_N$ are to be assigned bit depths $B_1, \ldots, B_N$ (which are not necessarily optimum.) We now investigate how the quantization step size $D_n$ should be decided given bit depth $B_n$. In the round-to-nearest scheme (RTN, Figure 2, left), $D_n$ is always chosen such that the quantizer's $2^{B_n}$ steps just cover the entire range of weight values, and this step size halves as $B_n$ is increased by one. These criteria optimize step sizes when weights are distributed uniformly across a range and the objective is to minimize distortion in quantized weights.

For LLMs, the elements $\theta$ of a weight matrix typically exhibit a light-tailed distribution $p_\theta$ (normal or Laplace) (Zhao et al., 2019), which renders partitioning the entire weight range into $2^{B_n}$ equal steps sub-optimal especially at low bit depths (Cover & Thomas, 2006; Gersho & Gray, 1991). One alternative to the computationally expensive Lloyd–Max algorithm (Lloyd, 1982; Max, 1960) is companded quantization (Gray & Neuhoff, 1998), which applies a sigmoid transform to $\theta$ prior to uniform quantization to achieve finer quantization in regions of larger $p_\theta$ and coarser quantization in regions where $p_\theta$ is smaller; see Figure 2 (right). When the weights $\theta$ are Laplace-distributed with mean $\mu$ and variance $S^2$, an asymptotically optimal choice of sigmoid function is (Appendix C):

$$\sigma(\theta, S, \mu) = \frac{1 + \text{sgn}(\theta - \mu)}{2} \exp\left(-\frac{\sqrt{2}\, \text{abs}(\theta - \mu)}{3S}\right) \in (0, 1), \quad \theta \in (-\infty, \infty), \qquad (8)$$

that is, the normalized cubic root of the cumulative distribution function for a Laplace distribution of the same mean and variance. Companded weights $\theta^\sigma = \sigma(\theta, S, \mu)$ are then quantized uniformly

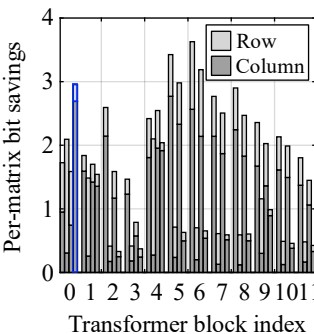 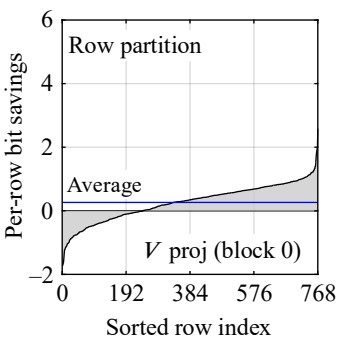 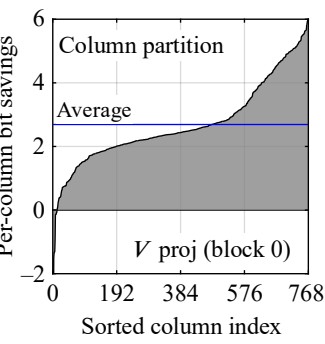

**Figure 3: Bit savings from partition.** Plotted for OPT-125m. Savings are derived by partitioning each weight matrix into a collection of row or column matrices and assigning each sub-matrix its own bit depth. Savings differ across the ($Q$, $K$, $V$ and $O$) projection matrices of the model's 12 transformer blocks (left). Per-column (middle) and row (right) bit savings (shown for block 3, $O$-proj) can dip below zero but are always positive on average due to Jensen's inequality (see text).

in the range $(0, 1)$ and signaled with bit depth $B$, scale $S$, and mean $\mu$ for efficient dequantization using lookup tables. In practice, $S, \mu$ are treated as hyper-parameters and fine-tuned efficiently on coarse 1D grids as a post-processing step (Young et al., 2021) once Algorithm 1 has completed.

Quantization invariably causes small deterministic differences to arise between the original (non-quantized) $\Theta$ and quantized $\Theta^q$ weights. While these errors are often modeled as zero-mean noise in theoretical analyses, they are seldom zero-mean in practice and can lead to systematically biased model output, which significantly reduces prediction accuracy. To compensate for these non-zero differences, we compute new bias vectors for the model as $\mathbf{B}_n^q \leftarrow \mathbf{B}_n + (\Theta_n^q - \Theta_n)\, \overline{\mathbf{X}}_n$ each time the matrix $\Theta_n$ is quantized. Here, $\overline{\mathbf{X}}_n$ is a vector of running means of the inputs to the $n$th layer, which is accumulated during the forward pass in a manner analogous to the accumulation of $G_n^2$ during the backward pass. The corrected biases $\mathbf{B}_n^q$ are then used whenever the corresponding quantized weight matrices $\Theta_n^q$ are used during gradient variance accumulation and inference.

## 3.3 MATRIX PARTITIONING

Rather than quantize optimally at the granularity of a whole weight matrix, we can split each matrix into a collection of row or column matrices, assigning optimum bit depth and step size to each sub-matrix. In this case, the total number of matrices $N$ in (3) can be reinterpreted as the total number of sub-matrices collected across all layers, with the quantities $B_n$, $D_n$ and $P_n$, similarly interpreted as the bit-depth, step size and number of elements of the $n$th sub-matrix. Note that quantizing at the granularity of row or column sub-matrices does not noticeably increase the complexity of variance accumulation, as the same squared gradients computed via back-propagation can be averaged per sub-matrix to produce the corresponding sub-matrix variances. Here, without loss of generality, we assume that each matrix is split into a collection of column matrices.

For a weight matrix $\Theta$ with gradient and weight variances $G^2$ and $S^2$, whose per-column variances are $G_1^2, \ldots, G_N^2$ and $S_1^2, \ldots, S_N^2$, respectively, the theoretical gain (average bit depth savings) from partitioning can be expressed as

$$\gamma_{\text{partition}} = \frac{1}{2}\left(\log_2\left(G^2 S^2\right) - \frac{1}{N}\sum_{n=1}^{N}\log_2\left(G_n^2 S_n^2\right)\right), \tag{9}$$

a non-negative quantity as a direct result of Jensen's inequality. This quantity represents the bit-rate (average bit-depth) savings when the $n$th column is assigned $B_n = \frac{1}{2}\log_2(G_n^2 S_n^2) + B$ bits for some $B$, compared to assigning a uniform bit depth $B_n = \frac{1}{2}\log_2(G^2 S^2) + B$ bits across all columns under the assumption that the weights of its $N$ columns are identically distributed. Figure 3 (left) plots the per-matrix bit-depth savings derived by partitioning the ($Q$, $K$, $V$ and $O$) projection matrices of the OPT-125m model by rows or columns. The per-channel breakdown of the savings is also shown.

In addition to primary splitting of matrices into columns, we may want to further split each column into a fixed number of groups of weight elements given the presence of row bit savings as well. To split the columns of a weight matrix $\Theta \in \mathbb{R}^{N \times N}$, one can simply cluster its rows into $M$ similarly

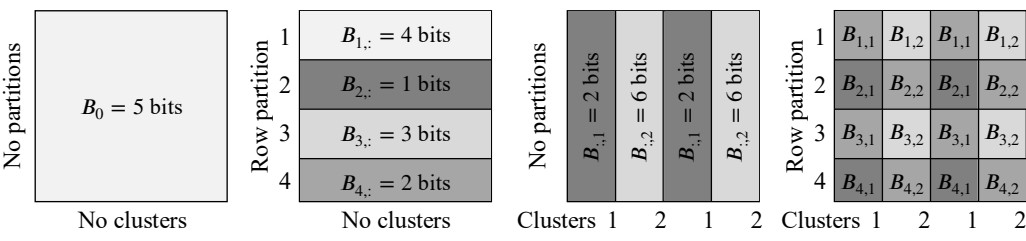

(a) No partitions or clusters    (b) Row partitioning    (c) Column clustering    (d) Partition and cluster

**Figure 4: Partitioning and clustering.** Illustrated for a $4 \times 4$ weight matrix. Rather than assign the same bit depth to all elements of a weight matrix (a), we can assign a separate bit depth to each row of weights (b), or to a cluster of columns (c), and even combine partitioning and clustering (d) to realize row- and column-based bit savings. Clustering with a cluster size of 2 illustrated for clarity.

sized groups based on their row variances $G_1^2 S_1^2, \ldots, G_N^2 S_N^2$. By applying the same clustering to all columns of a matrix, we can signal the cluster index for each row using $\lceil \log_2 M \rceil$ bits—a negligible per-weight overhead for a typical number of columns in a large weight matrix and number of groups used in practice. We illustrate partitioning and subdivision in Figure 4. Later in Section 4, we show the accuracy of OPT models quantized using different numbers of row clusters, demonstrating that the clustering idea used in AWQ and GPTQ similarly improves quantized model accuracy.

## 4   QUANTIZATION EXPERIMENTS

To study the behavior of quantized LLMs, we apply CVXQ (Algorithm 1) to the quantization of the Meta Open Pretrained Transformer (OPT) (S. Zhang et al., 2022) and Llama-2 (Touvron et al., 2023) families of language models (from the Hugging Face Hub), comparing the performance of CVXQ against other model quantization methods on language modeling and question answering tasks. For calibration data, we source 128 examples from the training split of the C4 dataset (Raffel et al., 2020) and test using the test split of WikiText2 (Merity et al., 2022) for language modeling and the test splits of GSM8K (Cobbe et al., 2021), ARC (Easy and Challenge) (Clark et al., 2018), HellaSwag (Zellers et al., 2019), PIQA (Bisk et al., 2019) and Winogrande (Sakaguchi et al., 2021) for question answering tasks.

**Language Modeling.** As our main set of experiments, we quantize Meta's OPT and Llama 2 models to 3 and 4 bits and measure the performance of the quantized models using perplexity, a stringent accuracy metric. We use row clustering with a cluster size of 512 for OPT (768 for 125M, 66B) and 256 for Llama 2 models, accumulation batch size of 16, and 17 tokens from each sequence of tokens of length 2048, and optimize for 64 iterations maximum. Table 1 lists the perplexity of our quantized models (CVXQ) on the WikiText2 test set. We select the final quantized model based on WikiText2 (validation) but selecting the last quantized model produces similar test accuracy (within 1% of the unquantized model's perplexity). For comparison, we include the perplexities of the same models quantized using round-to-nearest, GPTQ (Frantar et al., 2022), OWQ (Lee et al., 2024), AWQ (Lin et al., 2024) and QuIP (Chee et al., 2024) based on the code provided by the respective authors; see Appendix D for details. Relative to the next best performing methods, the proposed method provides a perplexity reduction of up to 4.55 for the 3-bit OPT-125M model but minor perplexity gains (0.00–0.01) are observed for the 3-bit OPT-66B and Llama 2 70B models. In this comparison, AWQ uses a group size of 128, incurring 2–4 times as many overhead bits as the proposed method, and OWQ by its nature operates at average per-weight bit depths that are 0.01–0.05 bits higher than proposed.

**Hyperparameters/Ablations.** To study the effect of CVXQ hyperparameters on the accuracy of the quantized models, we quantize the OPT-1.3B and -13B models over a range of minibatch sizes and token counts (optimization hyperparameters) and cluster sizes (quantization hyperparameter), with each hyperparameter varied while keeping the others fixed at their optimized values. (The optimal hyperparameter values are batch size: 16, token count: 17, and cluster size: 512.) The perplexity of the quantized models is then measured on the C4 test data. Table 2 (a–b) demonstrates that CVXQ is largely insensitive to the values of optimization hyperparameters over a wide range. From Table 2 (c), we see that smaller cluster sizes generally improve the performance of the quantized models at lower average bit depths, but this also leads to higher overheads (discussed later). Figure 5 plots quantized model accuracy across optimization iterations when the baseline hyperparameter values

**Table 1: WikiText2 perplexity (test).** We quantize the Meta OPT and Llama 2 families of LLMs to 3–4 bits per weight on average using the proposed quantization method, reporting the perplexity of each quantized model on the WikiText2 dataset (test). For comparison, we also include the perplexities of models quantized using RTN, GPTQ, QuIP, OWQ, and AWQ.

| Perplexity (PPL) WikiText2 (↓) | Meta OPT (Open Pretrained Transformer) | | | | | | | | Meta Llama 2 | | |
|---|---|---|---|---|---|---|---|---|---|---|---|
| | 125M | 350M | 1.3B | 2.7B | 6.7B | 13B | 30B | 66B | 7B | 13B | 70B |
| Full precision (FP16) | 27.65 | 22.00 | 14.63 | 12.47 | 10.86 | 10.13 | 9.56 | 9.34 | 5.47 | 4.88 | 3.32 |
| **4 bits** RTN | 37.28 | 25.94 | 48.17 | 16.92 | 12.10 | 11.32 | 10.98 | 111.36 | 5.73 | 4.98 | 3.46 |
| GPTQ | 32.05 | 23.87 | 15.47 | 12.83 | 11.14 | 10.29 | 9.57 | 9.34 | 6.07 | 5.20 | 3.59 |
| GPTQ/256 | 30.53 | 23.83 | 14.91 | 12.52 | 11.02 | 10.22 | 9.60 | 9.46 | 5.69 | 5.02 | 3.44 |
| QuIP | 35.93 | 23.15 | 15.96 | 12.67 | 11.10 | 10.33 | 9.60 | 9.40 | – | – | 3.53 |
| OWQ (4.01 bits) | 29.47 | 23.19 | 15.01 | 12.39 | 10.87 | 10.26 | 9.50 | 9.25 | 5.63 | 5.01 | 3.43 |
| AWQ/128 | 29.11 | – | 14.95 | 12.74 | 10.93 | 10.22 | 9.59 | 9.39 | 5.60 | **4.97** | 3.41 |
| CVXQ (Ours) | **27.23** | **22.89** | **14.20** | **12.12** | **10.52** | **10.08** | **9.45** | **9.13** | **5.57** | **4.97** | **3.40** |
| **3 bits** RTN | 1.3e3 | 64.57 | 119.47 | 298.00 | 23.54 | 46.04 | 18.80 | 6.1e3 | 6.66 | 5.52 | 3.98 |
| GPTQ | 53.43 | 32.28 | 20.90 | 16.55 | 12.88 | 11.58 | 10.29 | 9.90 | 9.23 | 6.69 | 3.87 |
| GPTQ/256 | 41.22 | 29.96 | 16.98 | 13.94 | 11.39 | 10.41 | 9.81 | 11.13 | 6.75 | 5.59 | 4.00 |
| QuIP | 34.43 | 26.02 | 17.33 | 13.84 | 12.35 | 10.57 | 9.92 | 9.46 | – | – | 3.85 |
| OWQ (3.01 bits) | 35.26 | 26.59 | 16.40 | 13.21 | 11.21 | 11.48 | 9.59 | 9.28 | 6.21 | 5.36 | 3.77 |
| AWQ/128 | 36.77 | – | 16.32 | 13.54 | 11.41 | 10.67 | 9.85 | 9.63 | 6.24 | 5.32 | 3.74 |
| CVXQ (Ours) | **30.71** | **25.94** | **14.83** | **12.42** | **11.07** | **10.28** | **9.56** | **9.24** | **6.04** | **5.25** | **3.72** |

**Table 2: Effect of hyperparameters on quantized model accuracy.** Quantized model accuracy is relatively insensitive to the minibatch size (a) and number of tokens per sequence (b) used for the optimization. Smaller clusters improve quantized model accuracy at low average bit depths (c). The gain from mixed precision and companding components are also shown in ablations (d).

(a) Minibatch size and PPL

| PPL C4 (↓) | OPT (4 bits) | | OPT (3 bits) | |
|---|---|---|---|---|
| | 1.3B | 13B | 1.3B | 13B |
| FP16 | 16.07 | 12.06 | 16.07 | 12.06 |
| 2 | 16.24 | 12.12 | 16.94 | 12.36 |
| 4 | 16.24 | 12.12 | 16.94 | 12.35 |
| 8 | 16.25 | 12.11 | 16.90 | 12.34 |
| **16** | **16.22** | **12.11** | **16.86** | **12.32** |
| 32 | 16.24 | 12.12 | 16.88 | 12.36 |

(Batch size)

(b) Number of tokens and PPL

| PPL C4 (↓) | OPT (4 bits) | | OPT (3 bits) | |
|---|---|---|---|---|
| | 1.3B | 13B | 1.3B | 13B |
| FP16 | 16.07 | 12.06 | 16.07 | 12.06 |
| 3 | 16.40 | 12.29 | 17.05 | 12.47 |
| 5 | 16.28 | 12.18 | 16.93 | 12.37 |
| 9 | 16.24 | 12.12 | 16.91 | 12.35 |
| **17** | **16.22** | **12.11** | **16.86** | **12.32** |
| 33 | 16.21 | 12.10 | 16.87 | 12.34 |

(Num tokens)

(c) Cluster size and PPL

| PPL C4 (↓) | OPT (4 bits) | | OPT (3 bits) | |
|---|---|---|---|---|
| | 1.3B | 13B | 1.3B | 13B |
| FP16 | 16.07 | 12.06 | 16.07 | 12.06 |
| 64 | 16.16 | 12.10 | 16.62 | 12.26 |
| 128 | 16.17 | 12.10 | 16.70 | 12.29 |
| 256 | 16.20 | 12.10 | 16.77 | 12.32 |
| **512** | **16.22** | **12.11** | **16.86** | **12.32** |
| 1024 | 16.23 | 12.11 | 16.99 | 12.42 |

(Cluster size)

(d) Ablation (Mixed prec. / Compand)

| PPL C4 (↓) | OPT (4 bits) | | OPT (3 bits) | |
|---|---|---|---|---|
| | 1.3B | 13B | 1.3B | 13B |
| FP16 | 16.07 | 12.06 | 16.07 | 12.06 |
| RTN | 24.51 | 13.36 | 4.2e3 | 3.2e3 |
| + MSE | 16.98 | 12.26 | 21.64 | 13.34 |
| + Mixed | 16.29 | 12.12 | 18.48 | 12.80 |
| + Comp | 16.22 | 12.11 | 16.86 | 12.32 |
| = Ours | **16.22** | **12.11** | **16.86** | **12.32** |

**Figure 5: Test perplexity across iterations.** Calibrated on C4 (train) using a batch size of 16. Row clusters of size 512 used. Perplexity decreases rapidly within the first 30 iterations, monotonically for C4 (test), whose distribution is similar to the calibration data.

are used, showing that about 20 iterations are needed for quantization parameters (clustering and bit depth decisions) to reach their optima. Table 2 (d) shows ablations of our quantized OPT models by starting with RTN and adding different components (Jeon et al., 2023). See Table 3 for C4 results.

**Pruning Due to Quantization.** CVXQ quantizes low-variance weights of weight matrices to zero and effects a form of weight pruning, which has been shown to improve generalization (Hassibi & Stork, 1992). Table 4 (a) lists the percentages of zero-quantized weights in the OPT-1.3B and 13B models quantized to 3 and 4 bits per weight on average. We observe that using smaller cluster sizes increases the number of pruned weights since this enables low-variance weights in each column to be clustered together and quantized to zero. However, smaller clusters lead to higher overheads so

**Table 3: C4 perplexity (validation).** We quantize the Meta OPT and Llama 2 families of LLMs to 3–4 bits per weight on average using the proposed quantization method, reporting the perplexity of each quantized model on the C4 dataset. For comparison, we also include the perplexities of models quantized using RTN, GPTQ, QuIP, OWQ, and AWQ.

| Perplexity (PPL) C4 ($\downarrow$) | Meta OPT (Open Pretrained Transformer) | | | | | | | | Meta Llama 2 | | |
|---|---|---|---|---|---|---|---|---|---|---|---|
| | 125M | 350M | 1.3B | 2.7B | 6.7B | 13B | 30B | 66B | 7B | 13B | 70B |
| Full precision (FP16) | 26.56 | 22.59 | 16.07 | 14.34 | 12.71 | 12.06 | 11.44 | 10.99 | 6.97 | 6.46 | 5.52 |
| RTN | 33.91 | 16.21 | 24.51 | 18.43 | 14.36 | 13.36 | 13.46 | 283.31 | 7.86 | 7.16 | 6.01 |
| GPTQ | 29.42 | 24.14 | 16.73 | 14.85 | 12.99 | 12.24 | 11.56 | 11.08 | 7.86 | 7.06 | 5.90 |
| GPTQ/256 | 28.36 | 24.18 | 16.47 | 14.64 | 12.88 | 12.15 | 11.50 | 11.12 | 7.58 | 6.88 | 5.79 |
| QuIP | 27.85 | 23.39 | 17.20 | 14.58 | 12.87 | 12.17 | 11.51 | 11.03 | – | – | 5.87 |
| OWQ (4.01 bits) | 27.93 | 23.37 | 16.49 | 14.60 | 12.83 | 12.17 | 11.49 | 11.02 | 7.59 | 6.94 | 5.81 |
| AWQ | 27.79 | – | 16.42 | 14.58 | 12.84 | 12.15 | 11.50 | 11.04 | 7.44 | 6.84 | 5.77 |
| CVXQ (Ours) | **27.27** | **23.20** | **16.24** | **14.44** | **12.79** | **12.11** | **11.48** | **11.01** | **7.40** | **6.83** | **5.76** |
| RTN | 839.97 | 55.96 | 4.2e3 | 1.1e4 | 4.4e3 | 3.2e3 | 1.1e3 | 3.5e3 | 521.22 | 14.01 | 11.06 |
| GPTQ | 42.64 | 29.90 | 20.46 | 17.48 | 14.56 | 13.16 | 12.14 | 11.53 | 11.44 | 8.98 | 7.12 |
| GPTQ/256 | 35.00 | 28.84 | 18.07 | 15.84 | 13.50 | 12.57 | 11.78 | 12.29 | 8.92 | 7.65 | 6.21 |
| QuIP | 31.37 | **25.58** | 18.15 | 15.92 | 13.66 | 12.40 | 11.67 | **11.16** | – | – | 6.14 |
| OWQ (3.01 bits) | 31.28 | 26.40 | 17.69 | 15.36 | 13.23 | 13.29 | 11.69 | 11.17 | 8.59 | 7.65 | 6.16 |
| AWQ | 32.91 | – | 17.81 | 15.49 | 13.34 | 12.55 | 11.75 | 11.26 | 8.30 | 7.31 | 6.04 |
| CVXQ (Ours) | **30.05** | 26.20 | **16.88** | **14.91** | **13.14** | **12.35** | **11.62** | 11.19 | **8.04** | **7.22** | **5.99** |

(4 bits for the first group of rows; 3 bits for the second group.)

that small improvements in generalization due to pruning come at the cost of signaling the overhead bits. Table 4 (b) lists the number of overhead bits (cluster indices and FP16 encodings of the location and scale parameters of each weight cluster) as a percentage of the total quantized weight bits. These overheads are in line with those of other algorithms which must similarly signal zero points and step sizes of the quantization grid (Lee et al., 2024).

**2.x-bit Llama-2.** We study the accuracy of Llama 2 models quantized to 2.x bits using CVXQ and OWQ, both of which are capable of quantizing models to fractional average bit depths. To enable a more comprehensive study, we compare against OWQ with no grouping, as well as with group sizes of 128 and 256. We see from Table 4 (a) that CVXQ-quantized Llama-2 models are considerably more accurate at these bit depths than their OWQ counterparts. This is expected since CVXQ assigns bit depths from the range $(0, B_{max})$ commensurately with gradient variances whereas OWQ opts to preserve the most sensitive (highest-variance) weights in FP16 and quantize the rest to 2 bits (Lee et al., 2024). In terms of execution time, CVXQ (64 iterations) and OWQ/GPTQ require 47 minutes and 18 minutes, respectively (excluding validation), to quantize the 7B model on Nvidia A100.

**Downstream Tasks (Common Sense QA, GSM8K).** To show the impact of model quantization on downstream tasks, we list in Table 5 (b) the accuracy of CVXQ-quantized Llama-2 models on the ARC (Clark et al., 2018), HellaSwag (Zellers et al., 2019), PIQA (Bisk et al., 2019) and Winogrande (Sakaguchi et al., 2021) common sense question answering, and GSM8K (Cobbe et al., 2021) math problem solving tasks. We set our cluster size and the group size of GPTQ and AWQ to 256. We observe that CVXQ produces slightly higher scores than the GPTQ and AWQ quantized 3-bit models while RTN leads to severely diminished scores despite having similar perplexity scores as CVXQ on WikiText2 (Table 1). We include example responses to GSM8K questions produced by different 3-bit quantized Llama-2-70B models in Appendix E.

## 5 DISCUSSION

Formulating weight quantization as a convex optimization problem, as we have done here, bestows several benefits. First, it explicates the objective we seek to optimize (minimizing output distortion in our case) and sets us on a path to solve the right problem using modern automatic differentiation tools e.g. PyTorch's autograd package. Second, our formulation enables us to interpret many earlier Hessian-based methods (Frantar et al., 2022; Lee et al., 2024; Dong et al., 2019; Chen et al., 2021) as heuristics for approximate optimization of the true underlying quantization objective. Note that (2) is a nonlinear system of equations in the bit depth variables, so that any non-iterative solution is necessarily only an approximate one if one's goal is to optimize an objective similar to (2). Recent high-performance model quantization methods (Chee et al., 2024; Frantar et al., 2022; Frantar &

**Table 4: Pruning and overhead bits.** A small fraction of weights is quantized to zero and pruned away due to low variance, with smaller clusters increasing the degree of pruning (a). Quantization incurs overhead bits for signaling cluster indices and location and scale parameters of clusters (b).

(a) Pruned columns (%) in quantized models

| Pruned (%) | OPT (4 bits) | | | OPT (3 bits) | | |
|---|---|---|---|---|---|---|
| Cluster size | 350M | 1.3B | 13B | 350M | 1.3B | 13B |
| 64 | 0.57 | 2.13 | 2.18 | 0.64 | 3.70 | 3.12 |
| 128 | 0.61 | 2.19 | 2.31 | 0.68 | 3.81 | 3.04 |
| 256 | 0.67 | 2.10 | 2.16 | 0.69 | 3.06 | 2.69 |
| **512** | **0.68** | **2.07** | **2.00** | **0.70** | **2.85** | **2.57** |
| 1024 | 0.68 | 2.08 | 1.92 | 0.70 | 2.39 | 2.26 |

(b) Overhead bits (%) from quantization parameters

| Overhead bits (%) | OPT (4 bits) | | | OPT (3 bits) | | | |
|---|---|---|---|---|---|---|---|
| Cluster size | 350M | 1.3B | 13B | 350M | 1.3B | 13B | 30B |
| 64 | 10.33 | 10.30 | 10.28 | 13.77 | 13.73 | 13.71 | 13.70 |
| 128 | 5.18 | 5.16 | 5.15 | 6.91 | 6.88 | 6.87 | 6.86 |
| 256 | 2.60 | 2.59 | 2.58 | 3.47 | 3.45 | 3.44 | 3.44 |
| **512** | **1.30** | **1.30** | **1.30** | **1.73** | **1.73** | **1.73** | **1.72** |
| 1024 | 0.64 | 0.65 | 0.65 | 0.85 | 0.87 | 0.87 | 0.86 |

**Table 5: 2.x-bit quantization and downstream tasks.** Quantized to 2.x bits per weight, CVXQ reduces perplexity considerably compared with OWQ models quantized to the same (a). Quantized model accuracy measured by performance on tasks such as GSM8K (b). Cluster size of 256 is used.

(a) Perplexity of 2.1–2.8 bit Llama 2 models

| Perplexity WikiText2 (↓) | Llama 2 7B (2.1–2.8 bits) | | | | |
|---|---|---|---|---|---|
| | 2.1 | 2.2 | 2.4 | 2.6 | 2.8 |
| FP16 | 5.47 | 5.47 | 5.47 | 5.47 | 5.47 |
| OWQ | 39.56 | 11.25 | 10.79 | 10.43 | 10.24 |
| OWQ/256 | 10.34 | 10.01 | 9.98 | 9.50 | 9.26 |
| OWQ/128 | 10.01 | 9.66 | 9.42 | 9.38 | 9.14 |
| CVXQ/256 | **9.47** | **8.39** | **7.05** | **6.56** | **6.21** |

(b) Scores for 3-bit Llama-2 models on GSM8K and QA

| Score (%) Llama-2 (↑) | GSM8K | | | Average QA | | |
|---|---|---|---|---|---|---|
| | 7B | 13B | 70B | 7B | 13B | 70B |
| FP16 | 64.83 | 67.82 | 72.36 | 14.10 | 23.43 | 53.90 |
| RTN | 1.82 | 1.67 | 6.14 | 39.32 | 52.15 | 58.22 |
| GPTQ/256 | 6.60 | 14.48 | 46.47 | 61.40 | 64.94 | 70.58 |
| AWQ/256 | 6.97 | 16.76 | 48.07 | 62.48 | 65.95 | 71.29 |
| CVXQ/256 | **7.81** | **18.20** | **49.81** | **62.82** | **66.37** | **71.87** |

(c) Scores for 3-bit Llama-2 models on common sense QA (Arc C, Arc E, HellaSwag, PIQA, Winogrande)

| Score (%) Llama-2 (↑) | Arc (Challenge) | | | Arc (Easy) | | | HellaSwag | | | PIQA | | | Winogrande | | |
|---|---|---|---|---|---|---|---|---|---|---|---|---|---|---|---|
| | 7B | 13B | 70B | 7B | 13B | 70B | 7B | 13B | 70B | 7B | 13B | 70B | 7B | 13B | 70B |
| FP16 | 43.34 | 48.38 | 54.27 | 76.30 | 79.42 | 82.74 | 57.13 | 60.04 | 64.76 | 78.07 | 79.05 | 82.15 | 69.30 | 72.22 | 77.90 |
| RTN | 20.73 | 30.63 | 37.20 | 34.97 | 60.65 | 65.66 | 31.09 | 43.73 | 51.01 | 57.34 | 70.40 | 73.12 | 52.49 | 55.33 | 64.09 |
| GPTQ/256 | 38.23 | 43.34 | 52.13 | 72.26 | 76.64 | 80.85 | 53.02 | 57.65 | 62.60 | 75.63 | 77.48 | 80.52 | 67.88 | 69.61 | 76.80 |
| QuIP | – | – | – | – | – | 79.31 | – | – | – | – | – | 80.25 | – | – | – |
| AWQ/256 | 41.13 | 45.05 | 53.24 | **73.36** | **77.95** | 81.69 | **54.06** | 57.83 | 63.64 | 75.84 | 77.26 | 81.66 | 68.03 | **71.67** | 76.24 |
| CVXQ/256 | **41.21** | **45.73** | **53.84** | 72.60 | **77.95** | **82.32** | 53.95 | **58.55** | **63.86** | **77.20** | **78.51** | **81.88** | **69.14** | 71.11 | **77.43** |

Alistarh, 2022; Lee et al., 2024) can ultimately trace their lineage back to the classic Optimal Brain Surgeon algorithm (Hassibi & Stork, 1992), which is a convex formulation for weight pruning, not quantization (see Appendix F). As a result, these methods inherit the need for fine-tuning as part of the quantization process, making them less suitable for the quantization of activations at inference time, where fine-tuning would lead to unacceptable delays in the inference pipeline.

Our experimental results indicate that an accurate characterization of the quantization problem can indeed lead to better compression outcomes. While the smaller OPT-125M model is too limited for practical use in many situations, its relative incompressibility helps contrast the performance of the different weight quantization methods themselves (Table 1). With larger models like OPT-66B and Llama 2-66B, most approaches (including RTN) perform similarly, suggesting that larger language models are more compressible in general. At first glance, RTN may seem sufficient for quantizing larger models. However, RTN-quantized models lead to severely reduced accuracy on downstream tasks such as GSM8K (Table 4 (a)), which highlights the importance of validating the accuracy of quantized models across multiple tasks and datasets (Jaiswal et al., 2024). Increasing the number of calibration examples (from 128 to 1024) did not noticeably affect the quantized model's perplexity on C4 ($\pm$ 0.01), which agrees with findings from previous reports (Hubara et al., 2021).

Our CUDA matmul kernel (Appendix A) accelerates matrix-vector multiplication by dequantizing mixed precision weights to floating point representation dynamically and multiplying with a floating point activation vector. For the $12288 \times 49152$ weight matrix of OPT-175B at 3 bits per weight on average, our kernel provides a 3.8x speed up over FP16 matrix-vector multiplication using cuBLAS matmul on Nvidia A6000. Accelerated matrix-vector multiplication, along with our low complexity quantization approach (once the optimal bit depths have been determined) allows us to apply CVXQ also to activation quantization, where quantization efficiency becomes paramount. Joint quantization of activation and weights is discussed in Part 2.

## REPRODUCIBILITY STATEMENT

To ensure the reproducibility of results in this work, we make our PyTorch CVXQ code available on our GitHub project website, where readers can also ask questions about this work. Appendix A lists our CUDA kernel. Appendices B–C provide derivations for our main theoretical results and Appendix D additionally details the code and command line options used to obtain the results of GPTQ (Frantar et al., 2022), OWQ (Lee et al., 2024), and AWQ (Lin et al., 2024).

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

## A    CVXQ KERNEL FOR MATRIX–VECTOR MULTIPLICATION

For completeness, we provide here a reference implementation for multiplication between a mixed-precision quantized matrix and full-precision vector multiplication. Here, we assign a single bit depth to each cluster of 4 rows, leading to e.g. 12288 different bit depths in the case of the $49152 \times 12288$ weight matrix (MLP layer) of the OPT-175B model. Consider a thread block size of $256 \times 256$. Each of $256 \times 256$ block, in turn, entails $1 \times 256$ threads, with each thread dequantizing a $256 \times 1$ column of weights and multiplying them with the matching $1 \times 256$ segment of a vector input. Bit depth changes every 4 rows but every thread will go through the same bit depth change in the course of multiplication, allowing divergence-free (and uniform memory access) operations.

```
__constant__ float lutable[256] = { DEQUANT }; // dequantized values
defined in macros.h

template <typename scalar_t>
__global__ void VecQuant3MatMulKernel(
    const   scalar_t* __restrict__ vec,
    const        int* __restrict__ mat,
             scalar_t* __restrict__ mul,
    const   uint8_t* __restrict__ depths,
    const   scalar_t* __restrict__ scales,
```

```
702      const      int* __restrict__ i_s,
703      const   uint8_t* __restrict__ shifts,
704
705      int height,
706      int width) {
707    int row = BLOCKHEIGHT * blockIdx.x;
708    int col =  BLOCKWIDTH * blockIdx.y + threadIdx.x;
709
710    __shared__ scalar_t blockvec[BLOCKWIDTH];
711    __shared__ scalar_t lut[BLOCKWIDTH];
712
713    blockvec[threadIdx.x] = scales[threadIdx.x / 4] * vec[(row / BLOCKHEIGHT)
     * BLOCKWIDTH + threadIdx.x];
714    lut[threadIdx.x] = lutable[threadIdx.x];
715    __syncthreads();
716
717    scalar_t res = 0;
718    int i = i_s[blockIdx.x * gridDim.y + blockIdx.y] + threadIdx.x;
719    // int i = width * row + col;
720    int shift = shifts[blockIdx.x * gridDim.y + blockIdx.y];
721
722    uint64_t tmp_curr;
723    uint32_t tmp_read;
724    uint32_t depth_;
725
726    int j = 0, k = 0;
727
728    tmp_read = reinterpret_cast<const uint32_t*>(mat)[i];
729    tmp_curr = static_cast<uint64_t>(tmp_read) << 32;
730    shift += 32;
731    i += width;
732
733    while (k < BLOCKWIDTH) {
734      depth_ = reinterpret_cast<const uint32_t*>(depths)[j];
735
736      int depth, bmask;
737      uint32_t index;
738      scalar_t szero, *table;
739      for (int d = 0; d < 32; d += 8) { // for each of the 4 depth clusters
     (represented in 8 bits)
740        depth = (depth_ >> (d + 0)) &  7;
741        bmask = (1 << depth) - 1;
742
743        szero = (static_cast<int>((depth_ >> (d + 3)) & 31) - 16) * 0.03125f;
744        table = reinterpret_cast<scalar_t*>(lut + (1 << depth));
745
746        if (shift + 4 * depth > 64) { // will run out of bits, read more
747          tmp_read = reinterpret_cast<const uint32_t*>(mat)[i];
748          tmp_curr   =    static_cast<uint64_t>(tmp_read)    <<    32    |
     static_cast<uint64_t>(tmp_curr) >> 32;
749          shift -= 32;
750          i += width;
751        }
752        index = (static_cast<uint32_t>(tmp_curr >> shift) & bmask);
753        res += blockvec[k + 0] * (szero + table[index]);
754        shift += depth;
755        index = (static_cast<uint32_t>(tmp_curr >> shift) & bmask);
756        res += blockvec[k + 1] * (szero + table[index]);
757        shift += depth;
758        index = (static_cast<uint32_t>(tmp_curr >> shift) & bmask);
759        res += blockvec[k + 2] * (szero + table[index]);
760        shift += depth;
761        index = (static_cast<uint32_t>(tmp_curr >> shift) & bmask);
```

```
756        res += blockvec[k + 3] * (szero + table[index]);
757        shift += depth;
758
759        k += 4;
760      }
761      j += 1;
762    }
762  atomicAdd(&mul[col], res);
763  }
```

## B  DERIVATION OF EQUATION (5)

To derive our main equation (5), we appeal to a linearized relationship between model weights and output, as well as standard results from rate–distortion theory (Gersho & Gray, 1991) that relate the quantization error of a random source to output distortion at a high bit depth, where the linearized model relationship is a good approximation. Let us start with our quantization objective

$$d(B_1, B_2, \ldots, B_N) = \mathbb{E}_{\mathbf{X}} \| f(\mathbf{X}, \mathbf{\Theta}_1^q(B_1), \mathbf{\Theta}_2^q(B_2), \ldots, \mathbf{\Theta}_N^q(B_N)) - f(\mathbf{X}) \|_F^2, \qquad (10)$$

in which $f(\mathbf{X}) = f(\mathbf{X}, \mathbf{\Theta}_1(B_1), \mathbf{\Theta}_2(B_2), \ldots, \mathbf{\Theta}_N(B_N))$ denotes the output of the unquantized model given input $\mathbf{X}$. We can write the residual and Jacobian of $f$ at $(\mathbf{X}, \mathbf{\Theta}_1^q(B_1), \mathbf{\Theta}_2^q(B_2), \ldots, \mathbf{\Theta}_N^q(B_N))$ as

$$r(\mathbf{X}, \mathbf{\Theta}_1^q, \mathbf{\Theta}_2^q, \ldots, \mathbf{\Theta}_N^q) = (r_1, \ldots, r_M)(\mathbf{X}, \mathbf{\Theta}_1^q, \mathbf{\Theta}_2^q, \ldots, \mathbf{\Theta}_N^q) = f(\mathbf{X}, \mathbf{\Theta}_1^q, \mathbf{\Theta}_2^q, \ldots, \mathbf{\Theta}_N^q) - f(\mathbf{X})$$

$$J(\mathbf{X}, \mathbf{\Theta}_1^q, \mathbf{\Theta}_2^q, \ldots, \mathbf{\Theta}_N^q) = \left( \frac{\partial f(\mathbf{X}, \mathbf{\Theta}_1^q, \ldots, \mathbf{\Theta}_N^q)}{\partial \mathbf{\Theta}_1}, \frac{\partial f(\mathbf{X}, \mathbf{\Theta}_1^q, \ldots, \mathbf{\Theta}_N^q)}{\partial \mathbf{\Theta}_2}, \ldots, \frac{\partial f(\mathbf{X}, \mathbf{\Theta}_1^q, \ldots, \mathbf{\Theta}_N^q)}{\partial \mathbf{\Theta}_N} \right) \qquad (11)$$

and proceed to write the gradient and Hessian of the objective (10) in terms of the $r$ and $J$ above as

$$\nabla d(\mathbf{X}, \mathbf{\Theta}_1^q, \mathbf{\Theta}_2^q, \ldots, \mathbf{\Theta}_N^q) = (J^T r)(\mathbf{X}, \mathbf{\Theta}_1^q, \mathbf{\Theta}_2^q, \ldots, \mathbf{\Theta}_N^q)$$

$$\nabla^2 d(\mathbf{X}, \mathbf{\Theta}_1^q, \mathbf{\Theta}_2^q, \ldots, \mathbf{\Theta}_N^q) = (J^T J)(\mathbf{X}, \mathbf{\Theta}_1^q, \mathbf{\Theta}_2^q, \ldots, \mathbf{\Theta}_N^q) + \underbrace{\sum_{m=1}^{M} (r_m \nabla^2 r_m)(\mathbf{X}, \mathbf{\Theta}_1^q, \mathbf{\Theta}_2^q, \ldots, \mathbf{\Theta}_N^q)}_{\approx 0} \qquad (12)$$

in which the second term of $\nabla^2 f$ is approximately zero either because the residuals $r_m$ are relatively small, or they are close to affine in $(\mathbf{\Delta}_1^q, \mathbf{\Delta}_2^q, \ldots, \mathbf{\Delta}_N^q)$ so that $\nabla^2 r_m$ are relatively small, which is the case in the vicinity of the solution.

Using (12), we can now express the local quadratic approximation of (10) about $(B_1, \ldots, B_N)$ as

$$\hat{d}(B_1, \ldots, B_N) \stackrel{(a)}{=} \mathbb{E}_{\mathbf{X}} \left[ \left( \mathbf{\Delta}_1^q(B_1), \ldots, \mathbf{\Delta}_N^q(B_N) \right) \left( (J^T J)(\mathbf{X}, \mathbf{\Theta}_1^q, \ldots, \mathbf{\Theta}_N^q) \right) \left( \mathbf{\Delta}_1^q(B_1), \ldots, \mathbf{\Delta}_N^q(B_N) \right)^T \right]$$

$$+ \underbrace{\mathbb{E}_{\mathbf{X}} \left[ \left( \Delta_1(B_1), \ldots, \Delta_N(B_N) \right)^T \left( (J^T r)(\mathbf{X}, \mathbf{\Theta}_1^q, \mathbf{\Theta}_2^q, \ldots, \mathbf{\Theta}_N^q) \right) \right]}_{= 0} \qquad (13)$$

$$\stackrel{(b)}{=} \sum_{n=1}^{N} \mathbb{E}_{\mathbf{X}} [(J^T J)_{nn}(\mathbf{X}, \mathbf{\Theta}_1^q, \ldots, \mathbf{\Theta}_N^q)] \mathbb{E} [\Delta_n^2(B_n)] \stackrel{(c)}{=} \sum_{n=1}^{N} \underbrace{P_n G_n^2 H_n S_n^2 2^{-2B_n}}_{= d_n(B_n)}$$

in which the zero expectation of the linear term in (a) follows from the zero means of quantization errors $\Delta_1, \ldots, \Delta_N$, (b) follows from the uncorrelatedness of $\Delta_1, \ldots, \Delta_N$, and (c) follows from our definition of gradient variance $G_n^2 = P_n^{-1} \mathbb{E}_{\mathbf{X}} [(J^T J)_{nn}(\mathbf{X}, \mathbf{\Theta}_1^q, \ldots, \mathbf{\Theta}_N^q)]$ together with the result from rate–distortion theory (Gersho & Gray, 1991) that relates the variance of random quantization error $\mathbb{E} [\Delta_n^2(B_n)] = H_n S_n^2 2^{-2B_n}$ to the variance $S_n^2$ of the random source, and the coefficient $H_n$, and bit depth $B_n$ of quantization. Expression (5) for the partial derivatives of $d$ with respect to $B_n$ follows directly from the properties of the derivative of an exponential.

Since (10) is a non-linear least-squares objective and its gradient depends on the gradient variances $G_1^2, G_2^2, \ldots, G_N^2$, its minimization requires an iterative update of $\mathbf{\Theta}_1^q, \mathbf{\Theta}_2^q, \ldots, \mathbf{\Theta}_N^q$ via the choice of $B_1, B_2, \ldots, B_N$ and re-evaluation of the gradient variances $G_1^2, G_2^2, \ldots, G_N^2$ at $\mathbf{\Theta}_1^q, \mathbf{\Theta}_2^q, \ldots, \mathbf{\Theta}_N^q$. This is similar to the local Hessian evaluated by the Gauss–Newton method (Nocedal & Wright, 2006) every time the descent direction is re-computed. One can think of $G_1^2, G_2^2, \ldots, G_N^2$ as the diagonal elements of a non-diagonal Hessian matrix used in e.g. the Gauss–Newton method, but whose off-diagonal elements disappear in the expectation due to multiplication by uncorrelated quantization errors $\mathbf{\Delta}_1^q, \ldots, \mathbf{\Delta}_N^q$.

## C  DERIVATION OF EQUATION (8)

To derive our sigmoid companding function (8), we turn to results from rate–distortion theory that relate the mean square error of quantization of weights $\theta$ to the density $p_\theta$ of $\theta$ and the density $\lambda(\theta)$ of quantization levels, where $2^B \int_a^b \lambda(\theta) \, d\theta$ expresses the number of quantization levels of a $B$-bit

quantizer within any interval $[a, b]$. Writing $\Pi_i$ for the $i$th quantization cell and $\Pi(\theta)$ for the width of the cell containing $\theta$, we can write the mean square error of quantized weights as

$$\mathbb{E}|\theta - \theta^q|^2 = \sum_{i=1}^{2^B} \mathbb{P}[\theta \in \Pi_i] \, \mathbb{E}[|\theta - \theta_i^q|^2 \mid \theta \in \Pi_i]$$

$$\stackrel{(a)}{\approx} \sum_{i=1}^{2^B} \mathbb{P}[\theta \in \Pi_i] \frac{|\Pi_i|^2}{12} \stackrel{(b)}{\approx} \int p_\theta(\theta) \frac{\Pi^2(\theta)}{12} \mathrm{d}\theta \tag{14}$$

$$\stackrel{(c)}{\approx} \frac{1}{2^{2B}} \int p_\theta(\theta) \frac{\lambda^{-2}(\theta)}{12} \mathrm{d}\theta$$

in which (a) follows from our assumption that weight distribution is approximately uniform within each quantization cell, (b) follows from an integral approximation of the finite sum, and (c) follows from the relationship $2^B \lambda^{-1}(\theta) = \Pi(\theta)$, all of which hold approximately when $B$ is sufficiently large.

To find the density $\lambda$ of quantization levels that leads to the minimum quantization error when $\theta$ has density $p_\theta$, we use Hölder's inequality: $\int p_\theta^{1/3} \leq (\int p_\theta \lambda^{-2})^{1/3} (\int \lambda)^{2/3}$. Since $\int \lambda = 1$, we have that $\int p_\theta \lambda^{-2} \geq (\int p_\theta^{1/3})^3$, which sets a lower bound on the last term of (14). This lower bound and hence minimum quantization error is attained iff $p_\theta \lambda^{-2} \propto \lambda$. The optimal density for quantization levels is therefore given by

$$\lambda(\theta) \propto p_\theta^{1/3}(\theta) \Longleftrightarrow \Pi^{-1}(\theta) \propto p_\theta^{1/3}(\theta). \tag{15}$$

Rather than optimize the density $\lambda$ to minimize the quantization error for a given $p_\theta$, we could equivalently transform the weights $\theta$ as $\theta^\sigma = \sigma(\theta)$ via a non-linear $\sigma$, so that uniform quantization applied to $\theta^\sigma \sim p_{\theta^\sigma}$ leads to the same minimum quantization error. The width $\Pi(\theta)$ of non-uniform quantization cells quantizing $\theta$ relates to the width of uniform quantization cells of the companded (transformed) weights $\theta^\sigma = \sigma(\theta)$ as

$$\mathrm{d}\sigma(\theta) = \frac{\mathrm{d}\theta}{\Pi(\theta)} \propto p_\theta^{1/3}(\theta) \mathrm{d}\theta \Longrightarrow \sigma'(\theta) \propto p_\theta^{1/3}(\theta), \tag{16}$$

in which the first proportionality follows from (15). We can find the optimal nonlinear transform $\sigma$ by integrating $p_\theta^{1/3}(\theta)$ and normalizing (for convenience) the range of the integral to $[0, 1]$:

$$\sigma(\theta) = \left( \int_{-\infty}^{\infty} p_\theta^{1/3}(t) \, \mathrm{d}t \right)^{-1} \left( \int_{-\infty}^{\theta} p_\theta^{1/3}(t) \, \mathrm{d}t \right) \tag{17}$$

(Gersho & Gray, 1991). Finally, we obtain (8) by substituting the expression for the density of a Laplace distribution (parameterized by mean $\mu$ and standard deviation $S$) into $p_\theta$ above. Transform $\sigma$ is asymptotically optimal as $B \to \infty$ in (14).

## D    ALGORITHM PARAMETERS

To aid the reproducibility of the results in Table 1, we document the code we used for all algorithms (RTN, GPTQ, OWQ, and AWQ) along with the command line arguments.

**RTN.** We use the OWQ code from `https://github.com/xvyaward/owq/tree/03cfc99` in the provided `owq` conda environment. In the case of e.g. Llama-2-7b-hf quantized to 3 bits, we run `python main.py meta-llama/Llama-2-7b-hf c4 --wbits 3 --nearest`.

**GPTQ.** We use the OWQ code from `https://github.com/xvyaward/owq/tree/03cfc99` in the provided `owq` conda environment. In the case of e.g. Llama-2-7b-hf quantized to 3 bits, we run the provided command `python main.py meta-llama/Llama-2-7b-hf c4 --wbits 3`. For results based on the group size of 256, we run `python main.py meta-llama/Llama-2-7b-hf c4 --wbits 3 –groupsize 256`.

**OWQ.** We use the OWQ code from `https://github.com/xvyaward/owq/tree/03cfc99` in the provided `owq` conda environment. In the case of e.g. Llama-2-7b-hf quantized to 3.01 bits, we

run the provided command `python main.py meta-llama/Llama-7b-hf c4 --wbits 3 -- target_bit 3.01`.

**AWQ.** We use the AWQ code `https://github.com/mit-han-lab/llm-awq/tree/3665e1a` in the provided `awq` conda environment. In the case of e.g. Llama-2-7b-hf quantized to 3 bits, we run the provided command `python -m awq.entry -model_path meta-llama/Llama-7b-hf --w_bit 3 --q_group_size 128 --run_awq --tasks wikitext`.

# E  OUTPUT PRODUCED BY DIFFERENT QUANTIZED MODELS

Table 5 lists output produced by different quantized Llama-2-70b models in response to questions taken from the GSM8K dataset. For each question, a prompt is created by prepending the question text with five other question and target pairs from the dataset (known as a 5-shot evaluation). This allows the model to establish a context for the required output and format. It is interesting to note that severe quantization errors (as in the case of RTN) manifest as non sequiturs and errors in logic rather than unintelligible output.

# F  CONVEX WEIGHT PRUNING (HASSIBI & STORK, 1992)

To facilitate comparison between convex weight quantization (this work) and the convex weight pruning work of Hassibi & Stork (1992), we provide a derivation of Hassibi & Stork's Optimum Brain Surgeon (OBS) algorithm (presented slightly differently), together with our commentary for additional clarification.

For simplicity, let us rewrite model (4) as $f(\,\cdot\,, \boldsymbol{\Theta}_1, \boldsymbol{\Theta}_2, \ldots, \boldsymbol{\Theta}_N) = f(\,\cdot\,, \boldsymbol{\Theta})$, where $\boldsymbol{\Theta}$ is a vector of all model weights across different layers of the model. The objective of convex weight pruning is to set some number of elements of $\boldsymbol{\Theta}$ to zero while fine-tuning the remaining elements to minimize the difference between the output of the pruned model $f(\,\cdot\,, \boldsymbol{\Theta}^p)$ and the output of the unpruned model $f(\,\cdot\,, \boldsymbol{\Theta})$. Writing the pruned weights as $\boldsymbol{\Theta}^p = \boldsymbol{\Theta} + \boldsymbol{\Delta}^p$, where $\boldsymbol{\Delta}^p$ is a vector of updates to be made to weights $\boldsymbol{\Theta}$, it is apparent that $\Delta_i^p = -\theta_i$ if the $i$th weight is to be pruned, otherwise $\Delta_i^p$ should be chosen to maximally compensate for the effect of other pruned weights on the output. Suppose we have decided to prune the $p$th element of $\boldsymbol{\Theta}$. The updated set of weights $\boldsymbol{\Theta}^p$ can be found by solving

$$\text{minimize} \quad d(\boldsymbol{\Delta}^p) = \mathbb{E}_{\mathbf{X}} \| f(\mathbf{X}, \boldsymbol{\Theta} + \boldsymbol{\Delta}^p) - f(\mathbf{X}) \|_2^2 \approx \mathbb{E}_{\mathbf{X}} \big[ \boldsymbol{\Delta}^{pT} (J^T J)(\mathbf{X}, \boldsymbol{\Theta}) \boldsymbol{\Delta}^p \big]$$
$$\text{subject to} \quad r(\boldsymbol{\Delta}^p) = \mathbf{e}_p^T \boldsymbol{\Delta}^p - \theta_p = 0 \tag{18}$$

in which $J(\mathbf{X}, \boldsymbol{\Theta})$ represents the Jacobian of $f(\mathbf{X}, \boldsymbol{\Theta})$ with respect to $\boldsymbol{\Theta}$, and $\mathbf{e}_p^T$ is an operator that picks out the $p$th element of a vector. The Lagrangian of this problem becomes

$$\mathscr{L}(\boldsymbol{\Delta}^p, \lambda) = \frac{1}{2} \mathbb{E}_{\mathbf{X}} \big[ \boldsymbol{\Delta}^{pT} (J^T J)(\mathbf{X}, \boldsymbol{\Theta}) \boldsymbol{\Delta}^p \big] + \lambda (\mathbf{e}_p^T \boldsymbol{\Delta}^p - \theta_p) \tag{19}$$

in which $\lambda$ represents the dual variable associated with the equality constraint $\mathbf{e}_p^T \boldsymbol{\Delta}^p - \theta_p = 0$.

To solve (18), we differentiate $\mathscr{L}$ with respect to $\boldsymbol{\Delta}^p, \lambda$ and set all obtained derivatives equal to 0 to obtain the first-order optimality conditions $\mathbb{E}_{\mathbf{X}} \big[ (J^T J)(\mathbf{X}, \boldsymbol{\Theta}) \big] \boldsymbol{\Delta}^p + \mathbf{e}_p \lambda = \mathbf{0}$ and $\mathbf{e}_p^T \boldsymbol{\Delta}^p - \theta_p = 0$. After some algebraic manipulations, we obtain the optimizing values

$$\boldsymbol{\Delta}^p = -\mathbb{E}_{\mathbf{X}} \big[ (J^T J)(\mathbf{X}, \boldsymbol{\Theta}) \big]^{-1} \mathbf{e}_p \lambda, \qquad \lambda = -\frac{\theta_p}{\mathbb{E}_{\mathbf{X}} \big[ (J^T J)(\mathbf{X}, \boldsymbol{\Theta}) \big]_{pp}^{-1}}, \tag{20}$$

in which the expression for $\lambda$ is obtained by substituting the expression for $\boldsymbol{\Delta}^p$ above into the second optimality condition $\mathbf{e}_p^T \boldsymbol{\Delta}^p - \theta_p = 0$ and solving for $\lambda$. Combining both expressions finally produces an update $\boldsymbol{\Delta}^p$ that minimizes the objective in (18):

$$\boldsymbol{\Delta}^p = -\frac{\theta_p}{\mathbb{E}_{\mathbf{X}} \big[ (J^T J)(\mathbf{X}, \boldsymbol{\Theta}) \big]_{pp}^{-1}} \mathbb{E}_{\mathbf{X}} \big[ (J^T J)(\mathbf{X}, \boldsymbol{\Theta}) \big]^{-1} \mathbf{e}_p, \qquad d(\boldsymbol{\Delta}^p) = \frac{1}{2} \frac{\theta_p^2}{\mathbb{E}_{\mathbf{X}} \big[ (J^T J)(\mathbf{X}, \boldsymbol{\Theta}) \big]_{pp}^{-1}}. \tag{21}$$

So far, we assumed that we were given the index $p$ of the weight to prune from $\boldsymbol{\Theta}$. To actually pick the best weights to prune away, we can compute the pruning loss $d(\boldsymbol{\Delta}^i)$ for all indices $i$, picking the

index $i$ associated with minimum loss. That is,

$$p = \arg\min_i \frac{1}{2} \frac{\theta_i^2}{\mathbb{E}_{\mathbf{X}}\left[(J^T J)(\mathbf{X}, \mathbf{\Theta})\right]_{ii}^{-1}},\tag{22}$$

after which $\mathbf{\Delta}^p$ can be calculated (and consequently $\mathbf{\Theta}^p$), and further weights to prune can be picked by initializing $\mathbf{\Theta} \leftarrow \mathbf{\Theta}^p$ and repeating the process until some pruning criterion has been met.

**Table 5: Output to questions from the GSM8K dataset (test) by 3-bit-quantized Llama-2 70B.** Scores computed across 1319 answers obtained using the flexible extract filter.

| Question | Target | FP16 (Score: 53.90%) | RTN (Score: 6.14%) | GPTQ (Score: 46.47%) | CVXQ (Score: 48.60%) |
|---|---|---|---|---|---|
| Question: Every day, Wendi feeds each of her chickens three cups of mixed chicken feed, containing seeds, mealworms and vegetables to help keep them healthy. She gives the chickens their feed in three separate meals. In the morning, she gives her flock of chickens 15 cups of feed. In the afternoon, she gives her chickens another 25 cups of feed. How many cups of feed does she need to give her chickens in the final meal of the day if the size of Wendi's flock is 20 chickens?\nAnswer: | If each chicken eats 3 cups of feed per day, then for 20 chickens they would need 3*20=<<3*20=60>>60 cups of feed per day.\nIf she feeds the flock 15 cups of feed in the morning, and 25 cups in the afternoon, then the final meal would require 60-15-25=<<60-15-25=20>>20 cups of chicken feed.\n#### 20 | Wendi feeds each chicken 3 cups of feed per day.\nShe feeds her flock 15 cups of feed in the morning and 25 cups of feed in the afternoon.\nThe total amount of feed she gives her flock is 15 cups + 25 cups = <<15+25=40>>40 cups.\nSince each chicken gets 3 cups of feed per day, the total number of chickens in her flock is 40 cups / 3 cups = <<40/3=13.33333333333333 3333333333333333333333 3333333333333333333333 3333333333333333333333 3333333333333333333333 3333333333333333333333 3333333333333333333 | 15 cups + 25 cups + 10 cups = <<15+25+10=50>>50 cups of feed.\n#### 50\n | 15 cups in the morning + 25 cups in the afternoon = <<15+25=40>>40 cups of feed\nEach chicken gets 3 cups of feed, so 40 cups of feed / 3 cups per chicken = <<40/3=13.33>>13.33 chickens\nSince there are 20 chickens, the number of cups of feed in the final meal is 20 - 13.33 = <<20-13.33=6.67>>6.67 cups\n#### 6.67\n\n | 15 cups + 25 cups = <<15+25=40>>40 cups of feed\n20 chickens x 3 cups = <<20*3=60>>60 cups of feed\n60 cups - 40 cups = <<60-40=20>>20 cups of feed\n#### 20\n\n |
| Question: A robe takes 2 bolts of blue fiber and half that much white fiber. How many bolts in total does it take?\nAnswer: | It takes 2/2=<<2/2=1>>1 bolt of white fiber\nSo the total amount of fabric is 2+1=<<2+1=3>>3 bolts of fabric\n#### 3 | 2 + 2/2 = <<2+2/2=3>>3 bolts\n#### 3\n\n | 2 blue bolts + 2 white bolts = <<2+2=4>>4 bolts in total.\n#### 4\np | 2 + 1 = <<2+1=3>>3 bolts\n#### 3\n\n | 2 + 2/2 = <<2+2/2=3>>3 bolts\n#### 3\n\n |
| Question: Josh decides to try flipping a house. He buys a house for $80,000 and then puts in $50,000 in repairs. This increased the value of the house by 150%. How much profit did he make?\nAnswer: | The cost of the house and repairs came out to 80,000+50,000=$<<80000+50000=130000>>130,000\nHe increased the value of the house by 80,000*1.5=<<80000*1.5=120000>>120,000\nSo the new value of the house is 120,000+80,000=$<<120000+80000=200000>>200,000\nSo he made a profit of 200,000-130,000=$<<200000-130000=70000>>70,000\n#### 70000 | The house is now worth 150% of $80,000, which is $80,000 * 150/100 = <<80000*150/100=120000>>$120,000.\nSo he made $120,000 - $80,000 - $50,000 = <<120000-80000-50000=20000>>$20,000.\n#### 20000\n\n | Josh bought the house for $80,000 and spent $50,000 on repairs. So he spent $80,000 + $50,000 = $<<80,000+50,000=130,000>>$130,000 on the house.\nSo Josh spent $130,000 on the house.\n#### $130,000\n | Josh bought the house for $80,000 and put in $50,000 in repairs.\nThe value of the house increased by 150%, so the new value is 1.5*$80,000+$50,000 = $<<1.5*80,000+50,000=180,000>>180,000.\nSo Josh made $180,000 - $80,000 - $50,000 = $<<180,000-80,000-50,000=50,000>>50,000 in profit.\n#### 50,000\n\n | The value of the house increased by 150%, so it's now 150% of $80,000, which is $120,000.\nJosh spent $80,000 + $50,000 = <<80000+50000=130000>>$130,000 on the house, so he made $120,000 - $130,000 = <<120000-130000=-10000>>-$10,000.\n#### -10000\n\n |

| | | | | | |
|---|---|---|---|---|---|
| Kylar went to the store to buy glasses for his new apartment. One glass costs $5, but every second glass costs only 60% of the price. Kylar wants to buy 16 glasses. How much does he need to pay for them?", "answer | The discount price of one glass is 60/100 * 5 = $<<60/100*5=3>>3.\nIf every second glass is cheaper, that means Kylar is going to buy 16 / 2 = <<16/2=8>>8 cheaper glasses.\nSo for the cheaper glasses, Kylar is going to pay 8 * 3 = $<<8*3=24>>24.\nAnd for the regular-priced glasses, Kylar will pay 8 * 5 = $<<8*5=40>>40.\nSo in total Kylar needs to pay 24 + 40 = $<<24+40=64>>64 for the glasses he wants to buy.\n#### 64 | The first glass costs $5.\nThe second glass costs $5 * 60% = $<<5*60%=3>>3.\nThe third glass costs $5.\nThe fourth glass costs $5 * 60% = $<<5*60%=3>>3.\nThe fifth glass costs $5.\nThe sixth glass costs $5 * 60% = $<<5*60%=3>>3.\nThe seventh glass costs $5.\nThe eighth glass costs $5 * 60% = $<<5*60%=3>>3.\nThe ninth glass costs $5.\nThe tenth glass costs $5 * 60% = $<<5*60%=3>>3.\nThe eleventh glass costs $5.\nThe twelfth glass costs $5 * 60% = $<<5*60%=3>>3.\nThe thirteenth glass costs $5.\nThe fourteenth glass costs $5 * 60% = $<<5*60%=3>>3.\nThe fifteenth glass costs $5.\nThe sixteenth glass costs $5 * 60% | The cost of one glass is $5, and the cost of the second glass is $5 * $<<5*2=10>>10>>10 is the cost of the third glass.\nThe cost of the fourth glass is $5 * $<<5*3=15>>15 is the cost of the fifth glass.\nThe cost of the sixth glass is $5 * $<<5*4=20>>20 is the cost of the seventh glass.\nThe cost of the eighth glass is $5 * $<<5*5=25>>25 is the cost of the ninth glass.\nThe cost of the tenth glass is $5 * $<<5*6=30>>30 is the cost of the eleventh glass.\nThe cost of the twelfth glass is $5 * $<<5*7=35>>35 is the cost of the thirteenth glass.\nThe cost of the fourteenth glass is $5 * $<<5*8=40>>40 is the cost of the fifteenth glass.\nThe cost of the sixteenth glass is $5 * $<<5*9=45>>4 | The first glass costs $5.\nThe second glass costs $5 * 60% = $<<5*60%=3>>3.\nThe third glass costs $5 * 60% * 60% = $<<5*60%*60%=1.8>>1.8.\nThe fourth glass costs $5 * 60% * 60% * 60% = $<<5*60%*60%*60%=1.08>>1.08.\nThe fifth glass costs $5 * 60% * 60% * 60% * 60% = $<<5*60%*60%*60%*60%=0.648>>0.648.\nThe sixth glass costs $5 * 60% * 60% * 60% * 60% = $<<5*60%*60%*60%*60%*60%=0.3888>>0.3888.\nThe seventh glass costs $5 * 6 | The first glass costs $5.\nThe second glass costs 60% of $5, which is $<<5*60%=3>>3.\nThe third glass costs 60% of $5, which is $<<5*60%=3>>3.\nThe fourth glass costs 60% of $5, which is $<<5*60%=3>>3.\nThe fifth glass costs 60% of $5, which is $<<5*60%=3>>3.\nThe sixth glass costs 60% of $5, which is $<<5*60%=3>>3.\nThe seventh glass costs 60% of $5, which is $<<5*60%=3>>3.\nThe eighth glass costs 60% of $5, which is $<<5*60%=3>>3.\nThe ninth glass costs 60% of $5, which is $<<5*60%=3>>3.\nThe tenth glass costs 60% of $5, which is $<<5*60%=3>>3.\nThe eleventh glass costs 60% |
| Question: Janet's ducks lay 16 eggs per day. She eats three for breakfast every morning and bakes muffins for her friends every day with four. She sells the remainder at the farmers' market daily for $2 per fresh duck egg. How much in dollars does she make every day at the farmers' market?\nAnswer: | Janet sells 16 - 3 - 4 = <<16-3-4=9>>9 duck eggs a day.\nShe makes 9 * 2 = $<<9*2=18>>18 every day at the farmer's market.\n#### 18 | Janet eats 3 eggs for breakfast and bakes 4 muffins, so she uses 3 + 4 = <<3+4=7>>7 eggs.\nShe sells the remainder, so she sells 16 - 7 = <<16-7=9>>9 eggs.\nShe sells each egg for $2, so she makes $2 * 9 = <<2*9=18>>18 dollars.\n#### 18\n\n | Janet sold 16 eggs per day. She sold three eggs for breakfast every morning, so she sold 16 - 3 = <<16-3=13>>13 eggs for breakfast.\nShe sold four eggs for her friends every day, so she sold 16 - 4 = <<16-4=12>>12 eggs for her friends.\nShe sold twelve eggs at the farmers' market daily, so she sold twelve eggs at the farmers' market daily, so she sold twelve eggs at the farmers' market daily, so she sold twelve eggs at the farmers' market daily, so she sold twelve eggs at | Janet eats 3 eggs and bakes 4, so she sells 16 - 3 - 4 = <<16-3-4=9>>9 eggs.\nShe sells 9 eggs for $2 each, so she makes $2 * 9 = <<2*9=18>>18 dollars.\n#### 18\n\n | Janet eats 3 eggs for breakfast and bakes 4 muffins, so she uses 3 + 4 = <<3+4=7>>7 eggs.\nShe sells the remainder at $2 per egg, so she makes $2 * (16 - 7) = <<2*(16-7)=2*9=18>>$18 per day.\n#### 18\n\n |

