# OpenReview forum: "LLM Compression with Convex Optimization—Part 1: Weight Quantization"
_ICLR.cc/2025/Conference — Submitted to ICLR 2025_

### Official Review · Reviewer_7R2x · 2024-10-20

**Soundness:** 2
**Presentation:** 2
**Contribution:** 2
**Rating:** 3
**Confidence:** 4

**Summary:**

This paper tackles the critical issue of large language model (LLM) compression, proposing a novel quantization technique, CVXQ, viewed from a convex optimization perspective. CVXQ, scalable to models with hundreds of billions of weight parameters, allows users to compress models to any specified size after training

**Strengths:**

The paper introduces a comprehensive quantization method for applying different bit allocation to groups within a large language model (LLM) matrix.

**Weaknesses:**

- The paper's contribution isn't distinct. Although it proposes treating dynamic bit allocation as a convex optimization problem, this approach faces several issues:
  - Mixed-precision quantization is a well-researched field; the paper should highlight how its method differs from existing techniques and why it chose to compare solely with LLM quantization methods.
  - Group quantization is not a new concept but a long-standing basic strategy in the quantization field.
  - The convex optimization formulation proposed seems flawed. For instance, in equation three, f(X) is not convex, which questions the validity of the entire problem.
- The writing needs improvement. The definition of "part-1" in the title is unclear, and many descriptions in the text are ambiguous.
- The utility of mixed precision within a matrix is unclear. This approach would require complex, specific hardware design, limiting its broad application. Most mixed-precision quantizations occur between layers, not within a matrix.

**Questions:**

See weaknesses.

---

> ### Author Response · Authors · 2024-11-13
>
> Apologies if my tone seems direct; it's meant purely for clarity.
> __________
> *1. "Mixed-precision quantization is a well-researched field ... highlight how its method differs from existing techniques ..."*
>
> While mixed-precision quantization has previously been explored (Wang et al., 2019; Chen et al., 2021; Lee et al., 2024; Dettmers et al., 2023), these methods assign different bit depths from a limited set of bit-depth options (e.g., 4 or 16 bits) or only across different layers. This is due to combinatorial nature of mixed bit-depth assignment. This limits the attainable quantized model accuracy especially for LLMs with hundreds of billions of parameters.
>
> In contrast, we formulate bit-depth assignment as a convex optimization problem. This allows us to overcome the combinatorial challenges faced by prior methods and to achieve true mixed-precision quantization at an arbitrary level of granularity (per-channel or per-layer) with a wider range of bit depth options, ({0, 1, 2, 3, 4, 5, 6, 7, 8}). This leads to optimal model quantization tailored specifically to the demands of each channel or layer.
>
> (These paragraphs now in the revised manuscript.)
> __________
> *"... why it chose to compare solely with LLM quantization methods ...*"
>
> Our paper is on LLM quantization. As such, we benchmark against state-of-the-art LLM quantization techniques, including mixed-precision and fixed-precision methods. These methods are also compared against in other LLM quantization works, ensuring a robust and contextually relevant evaluation of our method.
> __________
> *"2. ... Group quantization is not a new concept ...*"
>
> We meant to say that per-channel mixed precision works well with the grouping mechanism; see Table 2(c). The revised manuscript now correctly attributes the grouping mechanism to GPTQ and AWQ (Frantar et al., 2022; Lin et al., 2024).
> __________
> *"3.. ... The writing needs improvement. The definition of "part-1" in the title is unclear ..."*
>
> Thank you, we will revise/shorten the title per your suggestion.
> __________
> *"4. ... The convex optimization formulation proposed seems flawed. For instance, in equation three, f(X) is not convex ..."*
>
> __Not true.__  We never say or imply the network model $f$ is convex. It is the optimization objective $d$ that is convex with respect to continuous variables $B_1,\dots,B_N$. Objective $d$ is convex by construction since $f$ is linear(-ized) as in Hassibi and Stork (1992) and the MSE loss is convex. See eq. (5) and Appendix A for details.
> __________
> *"5. ... The utility of mixed precision within a matrix is unclear ... Most mixed-precision quantizations occur between layers, not within a matrix.*"
>
> Our work, as well as GPTQ, AWQ and OWQ (Frantar et al., 2022; Lin et al., 2024; Lee et al., 2024) are examples of weight-only quantization methods. None of these methods / their kernels perform arithmetic in 3 or 4 bits. At inference time, weights are de-quantized back into float16 so that they can multiply with float16 activations. This still amounts to acceleration because quantized weights can essentially travel faster through the memory hierarchy (registers–L1 cache–L2 cache–global). If weights must be dequantized, there is no need to insist upon mixed-precision quantization only across layers; see (Dettmers et al., 2022; Lee at al., 2024) for other examples of channel-wise mixed precision quantization. Our response to Pd64 clarifies this further.

---

> > ### Comment · Reviewer_7R2x · 2024-11-22
> > **Thanks for your expanation**
> >
> > Dear author,
> >
> > Thanks for your detailed explanation. I still have a question about your paper: What is the biggest gain through modeling the quantization problem as a convex problem instead of achieving some accuracy improvement (which seems less important since nowadays lots of methods can finetune the accuracy on a certain dataset)? I think making this problem more clearly would be helpful for this research.
> >
> > Thanks!

---

> ### Author Response · Authors · 2024-11-23
>
> *"What is the biggest gain through modeling the quantization problem as a convex problem"*
>
> Higher quantized model accuracy is not the focus of this paper (even if it is one of our findings). Our work shows that using mixed precision bit depths (determined using optimization) and with the classic round-to-nearest quantization can outperform more complex GPTQ-like methods. Our low complexity quantization operations (scaling and rounding) can even facilitate activation quantization at inference time. (Ln 59–62)
>
> Quantizing a matrix of activations (e.g., during batched inference) using GPTQ-like methods can take tens of seconds, introducing significant delays to inference. Essentially,  these methods are procedural, requiring the same computational effort for subsequent quantizations of similar activation matrices or even the same weight matrix (if there is a need to quantize the same weight matrix again the second time). (LN 45–47). This contrasts with our method where only scaling and rounding is needed for the actual quantization once the optimal bit depths have been obtained.
>
> In this work, we lay the theoretical groundwork for determining the optimal bit depths using optimization and demonstrate the effectiveness of this approach in the context of weight-only quantization, where the bulk of earlier work lies.

---

### Official Review · Reviewer_z2t1 · 2024-11-03

**Soundness:** 3
**Presentation:** 3
**Contribution:** 2
**Rating:** 3
**Confidence:** 5

**Summary:**

This paper presents a framework for efficient handling of large language models (LLMs) by (1) determining mixed-precision quantization at layer or group levels to meet a target bitwidth and (2) proposing a novel method for deciding quantization step sizes.

**Strengths:**

The proposed techniques are well-grounded in theory, and each aspect of the framework appears logically sound and justifiable.

**Weaknesses:**

The paper introduces a mixed-precision approach, but comparisons are primarily made with uniform-precision quantization methods. A broader survey and comparison with other mixed-precision methods, addressing their strengths and weaknesses, would provide a stronger context for evaluating the proposed method.

An ablation study is needed. According to Z-Fold [1], step size determination methods like Min-Max, MMSE, and Hessian-based approaches are often used in quantization. A comparative analysis showing the effectiveness of the proposed method against these would strengthen the evaluation.

Separating the processes of bit-precision allocation and the quantization algorithm applied could provide clearer insights into each aspect of the method.

The proposed methodology is reasonable but lacks comparative analysis, which would underscore its relative advantages.


Testing on a wider range of models and benchmarks would further validate the generalizability of the proposed approach.




[1] Jeon et al. "A frustratingly easy post-training quantization scheme for llms." Proceedings of the 2023 Conference on Empirical Methods in Natural Language Processing. 2023.

**Questions:**

The paper claims that the proposed algorithm completes the quantization quickly, yet a lack of experimental or theoretical analysis supports this assertion. Could the authors provide more evidence or discussion on this aspect?

---

> ### Author Response · Authors · 2024-11-24
>
> Apologies in advance if my tone seems direct; it's meant purely for clarity.
> ________________
> *"An ablation study is needed."*
>
> Revised manuscript now includes an ablation in Table 2 (d). We start with RTN (min-max), to which we add MSE, add mixed precision, and finally add companding to arrive at the proposed approach. At 4 bits, mixed precision bit depth assignment is the dominant component. At 3 bits, both mixed precision and companding play important roles. The work of [1] Jeon et al. is now cited on ln 425.
> ________________
> *"Separating the processes of bit-precision allocation and the quantization ..."*
>
> Indeed, Algorithm 1 should be called "git depth determination". Once the bit depths are obtained (along with the scale factors), the actual quantization is simply scaling the weights using the scale factor, and uniformly quantizing over [0, 1].
> ________________
> *"The proposed methodology is reasonable but lacks comparative analysis ... Testing on a wider range of models and benchmarks."*
>
> The revised manuscript includes more benchmarks: C4 (complexity), Arc Challenge, Arc Easy, HellaSwag, PIQA and Winogrande. We already test on 11 models of various sizes (125M – 70B) that our baseline methods also test on. Many baseline methods (GPTQ, QuIP, etc) do not not have official code available for non-OPT / non-Llama models.
> ________________
> *"The paper claims that the proposed algorithm completes the quantization quickly ... discussion on this aspect."*
>
> Once the optimal bit depths have been determined, we simply perform round-to-nearest quantization of weights scaled to the range [0, 1]. The revised manuscript now clarifies this.

---

> > ### Comment · Reviewer_z2t1 · 2024-11-27
> >
> > Thank you for your effort to address my concerns; however, my main concern as below still remains unresolved. It seems essential to address this issue in order for the work to be considered for publication
> >
> > "The paper introduces a mixed-precision approach, but comparisons are primarily made with uniform-precision quantization methods. A broader survey and comparison with other mixed-precision methods, addressing their strengths and weaknesses, would provide a stronger context for evaluating the proposed method."

---

> ### Author Response · Authors · 2024-11-27
>
> Thank you. Slim-LLM (https://openreview.net/forum?id=tjlTczcnPz) is a notable mixed precision method (which extends GPTQ). A potential drawback of implementing mixed precision (or any extension) on top of GPTQ is that the resulting method inherits the need to perform Cholesky decomposition during quantization, precluding the method from being applied to activation quantization (as Cholesky decomposition would need to be applied to activations at inference time). Unlike Slim-LLM, our method is a mixed-precision RTN scheme, side-stepping GPTQ's Cholesky decomposition altogether. We now make this distinction from Slim-LLM more clear in the revised manuscript. Since Slim-LLM is contemporaneous work, we were not required to compare our work against it (see https://iclr.cc/Conferences/2025/FAQ).

---

### Official Review · Reviewer_Pd64 · 2024-11-03

**Soundness:** 2
**Presentation:** 3
**Contribution:** 1
**Rating:** 3
**Confidence:** 4

**Summary:**

The authors introduce a quantization framework called CVXQ, which first optimizes bit depth assignment and then refines step sizes and biases using convex optimization techniques. To further improve the quantization scheme, the framework incorporates matrix partitioning, dividing the matrix into a set of row or column sub-matrices, each with its own bit depth and step size. The experiments are conducted on Meta's LLaMA and OPT models, using PPL and GSM8K as evaluation metrics.

**Strengths:**

The authors derive several mathematical formulations for the quantization scheme, making a few assumptions about weight distributions, such as Normal or Laplace. They use figures to illustrate whether the statistical data from the OPT models align with these assumed distributions.

**Weaknesses:**

The main concern with this manuscript is that it does not address practical hardware constraints. Specifically, the authors permit each weight to have a different bit depth assignment, a strategy that is rarely seen in existing literature. For instance, AWQ employs dedicated kernels and uniformly quantizes all weights to 4 bits, aligning with the availability of a 4-bit engine. However, the manuscript lacks discussion on hardware acceleration or performance degradation resulting from the proposed quantization scheme.

By neglecting hardware-related considerations, the comparisons with previous works may appear unfair. Well-established quantization methods like OWQ, AWQ, or RTN explicitly demonstrate how their quantized models achieve latency improvements on common GPUs. In contrast, this manuscript explores more complex ideas, such as pruning and matrix partitioning, without addressing the impact on parallelism or the hardware requirements these approaches would entail.

It is crucial to describe the limitations of the quantization scheme for practical hardware implementation. Without doing so, methods that account for hardware acceleration might seem inadequate, despite the practical challenges associated with mixed precision or varying bit depth assignments.

For example, the authors should clarify how different bit-depth assignments would affect matrix multiplication kernels as batch size increases, as this could have a significant impact on performance.

In summary, the major concerns are: 1) the lack of considerations for hardware acceleration; 2) the use of configurations, such as varying bit depths, that seem impractical and create unfair comparisons with prior work; and 3) the need for a reevaluation of experimental results, given that the proposed quantization schemes operate under fundamentally different assumptions.

**Questions:**

Please see weaknesses

---

> ### Author Response · Authors · 2024-11-13
>
> Apologies in advance if my tone seems direct; it's meant purely for clarity.
> _________________
> *1. "The main concern ... the authors permit each weight to have a different bit depth assignment, ..."*
>
> __Not true.__ We use a grouping/clustering idea similar to GPTQ and AWQ (Frantar et al., 2022; Lin et al., 2024). Ln 356 says that we assign a single bit depth to a group of 512 weights (OPT) or 256 weights (Llama-2). Table 2(c) also shows that a group size of 512 performs better than 64, 128, or 256 for our quantized OPT models. For added clarity, we add to the caption of Figure 4: ... Clustering with a cluster size of 2 illustrated only for clarity.
> _________________
> *2. "... AWQ employs dedicated kernels and uniformly quantizes all weights to 4 bits, aligning with the availability of a 4-bit engine. However, the manuscript lacks discussion on hardware acceleration or performance degradation."*
>
> __Not true.__ GPTQ and AWQ (Frantar et al., 2022; Lin et al., 2024) and their engines do not perform arithmetic in 4 or 3 bits. These methods dequantize 4- or 3-bit weights back into float16 on the fly so that weight-activation multiplication can performed in float16. We dequantize mixed precision weights (some of which are 3 bits, some 4 bits, some 8 bits, etc.) back into float16 in exactly the same way.
> __________________
> *3. "... clarify how different bit-depth assignments would affect matrix multiplication kernels as batch size increases, as this could have a significant impact on performance."*
>
> Increasing the batch size does not change inference, as weights are always dequantized back to float16 as a first step and activations are always kept in float16. Indeed, this is the approach used also used by (Frantar et al., 2022; Lin et al., 2024; Lee et al., 2024).
> __________________
> *4. "the lack of considerations for hardware acceleration"*
>
> __Not true.__ Our method leads to hardware acceleration in exactly the same manner as GPTQ, AWQ, and OWQ. For example, our kernel (see (the supplementary zip file) provides a 3.8x speed up for matrix vector multiplication on the 12288 x 49152 weight matrix (dense layer) of OPT-175B at 3 bits per weight on average, relative to FP16 matmul using cuBLAS (measured on A6000).
> __________________
> *5. "Use of configurations, such as varying bit depths, that seem impractical and create unfair comparisons with prior work ... the need for a reevaluation of experimental results, given that the proposed quantization schemes operate under fundamentally different assumptions"*
>
> __Not true.__ The methods you mention (GPTQ, AWQ and OWQ) do __not__ perform arithmetic directly in 3 or 4 bit weights and activations. They assume weights will be de-quantized back to float16 as needed for weight–activation multiplications in float16. This is the same assumption that our method is based on. Our method has the additional flexibility of assigning different bit depths (8, 7, 6, 5, 4, 3, 2, 1, or even 0 bits) to different groups of 512 weights to maximize the accuracy of the quantized model while maintaining the same de-quantization complexity (since everything is scalar quantized).

---

> > ### Comment · Reviewer_Pd64 · 2024-11-24
> > **Response to the authors**
> >
> > Unfortunately, it appears that the authors have not implemented dequantization kernels in practice. In the Nvidia ecosystem, dequantization is typically executed on CUDA cores, which offer a lot lower throughput compared to Tensor cores. As a result, significant efforts are being made by practitioners to reduce dequantization overhead on CUDA cores through innovative techniques. For instance, recent FP6 kernels illustrate how engineers are working to improve dequantization throughput, even for large batch sizes.
> >
> > The authors are strongly encouraged to study the practical advancements in recent dequantization kernels.
> >
> > While I maintain my original score, I am strongly inclined toward rejection.

---

> ### Author Response · Authors · 2024-11-24
>
> *"Unfortunately, it appears that the authors have not implemented dequantization kernels in practice."*
>
> __Not true.__ Our dequantization kernel is in in the new supplementary zip file. For e.g. the 12288 x 49152 weight matrix (dense layer) of OPT-175B at 3 bits per weight on average, our kernel provides a 3.8x speed up over FP16 matrix-vector multiplication using cuBLAS matmul (on A6000). This is also stated in the revised manuscript.

---

> > ### Comment · Reviewer_Pd64 · 2024-11-25
> > **Response to the authors**
> >
> > There is a significant lack of detail regarding the kernel design. What are the specific hardware choices considered for the kernel design? Is it A100, A6000, H100, or something else? Additionally, what is the expected performance across other hardware options? What are the limitations of the proposed kernels, and how does batch size impact performance?
> >
> > I recommend referring to other documents to see how kernel performance is typically demonstrated.
> >
> > I also strongly align with the feedback provided by the other reviewers.

---

> ### Author Response · Authors · 2024-11-25
>
> *"What are the specific hardware choices considered for the kernel design?"*
>
> Ln 635 (revised manuscript) says that we measured the 3.8x matmul speed-up on a A6000. Note Our submission focuses primarily on convex optimiazation for model quantization, not kernel design. While expected performance across other hardware options, while nice to discuss, is not essential to the mathematical framework being presented here.
> _______________
> *"I recommend referring to other documents to see how kernel performance is typically demonstrated.*"
>
> What "documents" are you referring to? References? For example, the baseline methods AWQ (AAAI), QuIP (NeurIPS) report kernel performance similarly on a single GPU model. Only AWQ (MLSys) explores more hardware options because because MLSys emphasizes hardware aspects of ML. Each venue has different expectations on what a model quantization paper should entail. In this ICLR submission, we focus more on the mathematical / rate–distortion-theoretic aspects of model quantization.

---

### Official Review · Reviewer_EzZu · 2024-11-04

**Soundness:** 2
**Presentation:** 2
**Contribution:** 2
**Rating:** 3
**Confidence:** 4

**Summary:**

This paper proposes a method called CVXQ for mixed precision weight-only quantization of large language models (LLMs) using convex optimization techniques. CVXQ allows for user-specific quantization bit depths by defining the average bit depth and then seeking to minimize quantization error within this constraint. The method introduces row-wise and column-wise clustering to achieve this goal, where each cluster can be assigned different bit depths. To assign these bit depths, the problem is formulated in a Lagrangian form and solved using convex optimization. The effectiveness of CVXQ is demonstrated by achieving superior performance on the WikiText perplexity (PPL) metric compared to methods such as GPTQ, AWQ, and OWQ across various sizes of OPT models.

**Strengths:**

* Demonstrates that companded quantization can reduce the mean square error of weights before and after quantization more effectively than uniform quantization.
* Introduces a novel approach to weight-only quantization by employing various partitioning methods, specifically row and column clustering.
* Proposes a method to minimize the degradation in performance due to quantization within a constrained average bit depth by finding the optimal bit assignment combination. This is achieved by defining the quantization objective function in a Lagrangian form and solving it using convex optimization.
* Shows that the proposed partitioning methods can result in greater bit depth savings compared to non-partitioned methods.

**Weaknesses:**

* Lacks comparison with existing LLM quantization methods such as FlexRound[1] and QuIP[2].
* Primarily evaluates LLM performance using perplexity, with insufficient comparison across other metrics like MMLU and AlpacaEval.
* Insufficient discussion and comparative analysis on how the proposed CVXQ method can be accelerated on existing hardware such as GPUs. One of the key goals of compression methods like quantization is to achieve actual acceleration. Although the paper mentions that this will be addressed in Part 2, it is crucial to include a discussion on how to accelerate the proposed quantization format.
* Tables 1 and 2 lack information on the average bit depth achieved by CVXQ. Since the proposed method assigns bit depths through a convex optimization process, it may not exactly match the user-specific quantization bit depth, leading to potentially different compression rates in practice.
* The quantization process described in the paper suggests that the time required for quantization might exponentially increase with the number of iterations, as shown in Figure 5.

[1] FlexRound: Learnable Rounding by Element-wise Division for Post-Training Quantization, https://openreview.net/forum?id=-tYCaP0phY_

[2] QuIP: 2-Bit Quantization of Large Language Models With Guarantees, https://arxiv.org/abs/2307.13304

**Questions:**

* What do the terms "row" and "column" mean in the context of row and column partitioning in Figure 3?
* What units were used for clustering in Tables 1 and 2?
* The Massive Activation paper[3] demonstrated significant performance degradation when clipping massive activations from activation distributions. Papers like LLM.int8, SmoothQuant, and AWQ have shown the importance of considering activation distributions to mitigate the impact of outliers. Can the proposed CVXQ method be extended to apply to activation distribution?
* How does the size of the calibration set affect the performance of CVXQ?

[3] Massive Activations in Large Language Models, https://arxiv.org/abs/2402.17762

---

> ### Author Response · Authors · 2024-11-13
>
> Apologies if my tone seems direct; it's meant purely for clarity.
> _________________________
> *"Lacks comparison with existing LLM quantization methods such as FlexRound [1] and QuIP [2]."*
>
> FlexRound does not have a publicly available code and its results are not reproducible. The revised manuscript lists QuIP results in Tables 1 and 3 (ln 398, 394, 441, 447). Note that the official QuIP code does not work correctly on Llama-2 models (a known issue reported on the QuIP GitHub), producing perplexities higher than RTN. Out of respect for the authors of QuIP, we do not report these QuIP results on Llama-2 models.
> _________________________
> *"Primarily evaluates LLM performance using perplexity, with insufficient comparison across other metrics like MMLU and AlpacaEval."*
>
> We have some concerns with MMLU and AlpacaEval, as they are not very widely used, not even in the FlexRound and QuIP works that you refer to. So, we include one more perplexity metric (on C4) as well as the following new QA metrics: Arc (Challenge), Arc (Easy), HellaSwag, PIQA, and Winogrande. These are popularly used in other model compression papers. The revised manuscript shows these results in the Table 5.
> __________________________
> *"Insufficient discussion ... accelerated on existing hardware such as GPUs."*
>
> Our custom CUDA kernel achieve 3.3–3.8x acceleration for matrix-vector multiplication. The revised manuscript discusses this (ln 41, 532) and lists the kernel code in Appendix A.
> __________________________
> *"Tables 1 and 2 lack information on the average bit depth achieved by CVXQ ... may not exactly match the user-specific quantization bit depth ..."*
>
> Our convergence tolerance is $10^{-6}$ bits (ln 233). The actual average bit depths achieved by CVXQ were 3.999999–4.000001 (for 4-bit models) and 2.999999–3.000001 (for 3 bit models).
> __________________________
> *"What do the terms "row" and "column" mean in the context of row and column partitioning in Figure 3?"*
>
> Row (resp. column) refers to the average bit depth savings achieved when assigning separate bit depths to rows (resp. columns) of each weight matrix. This is also stated more clearly in the revised manuscript.
> __________________________
> *"What units were used for clustering in Tables 1 and 2?"*
>
> Ln 356 states that cluster sizes of 512 (OPT) and 256 (Llama-2) are used.
> __________________________
> *"The Massive Activation paper[3] demonstrated significant performance degradation when clipping massive activations from activation distributions ... Can the proposed CVXQ method be extended to apply to activation distribution?*"
>
> CVXQ already considers activation distribution. CVXQ uses the mean square magnitude of the weight gradient to inform bit depth assignment. By expressing weight gradient as the outer product of the input to the weights and the gradient of weight's output, we see that input magnitudes are indeed considered in the form of (mean square) gradient magnitude.
> ___________________________
> *"The quantization process described in the paper suggests that the time required for quantization might exponentially increase with the number of iterations, as shown in Figure 5.*"
>
> __Not true.__ The number of iterations is kept the same across model sizes and the time each iteration takes is roughly linear (slightly super-linear) in the number of model parameters. CVXQ takes 47m to quantize the Llama-2-7B model, and 12h for the Llama-2-70B one. The revised manuscript now states these timings.
> ___________________________
> *"How does the size of the calibration set affect the performance of CVXQ?*"
>
> For OPT-1.3B and OPT-13B models, we experimented using 1024 calibration samples instead of 128 and the resulting perplexities on C4 were within ±0.01 of those based on 128 samples. This is consistent with the variance observed from choosing a different set of 128 calibration examples (ln 530).

---

> ### Comment · Reviewer_EzZu · 2024-11-25
> **Response to the authors**
>
> Thank you for the authors' response. However, I still feel that my questions have not been fully addressed. Specifically, if the kernel structure shows similar acceleration at the same compression ratio in both GPTQ and AWQ, the advantage of the proposed paper should be demonstrated through a better trade-off point between compression ratio and accuracy. However, I do not understand how the proposed method has been reasonably compared with existing methods such as FlexRound or QuIP. Additionally, I am skeptical that a lower perplexity score, such as Table 1 and 2, necessarily indicates better quality in LLMs. Therefore, I thinik the authors need to conduct a more rigorous comparison in terms of accuracy. For these reasons, I have decided to maintain my original score.

---

> ### Author Response · Authors · 2024-11-25
>
> *"From running QuIP and other methods, I've observed cases where lower perplexity scores resulted in significant degradation in MMLU performance."*
>
> Is your finding about MMLU vs perplexity published anywhere? Is it citable? None of our baseline methods report on MMLU. It is not good scholarly practice to make an anecdotal statement especially during a review.
>
> The revised manuscript includes QuIP results, as well as 6 more metrics (Scores on Arc Easy, Arc Challenge, HellaSwag, PIQA, Winogrande, C4 perplexity). This is in addition to WikiText2 and GSM8K. We believe this is sufficient validation from multiple angles, and is in line with the amount / kind of validation performed by others.

---

> ### Comment · Reviewer_EzZu · 2024-11-25
> **Response to the authors**
>
> > The revised manuscript includes QuIP results, as well as 6 more metrics (Scores on Arc Easy, Arc Challenge, HellaSwag, PIQA, Winogrande, C4 perplexity). This is in addition to WikiText2 and GSM8K. We believe this is sufficient validation from multiple angles, and is in line with the amount / kind of validation performed by others.
>
> Thank you for your response. I have reviewed the comparison results for QuIP. Could you please provide the Common Sense Reasoning (CSR) or MMLU results for the comparison between the proposed method and other state-of-the-art baselines such as QuIP?

---

> > ### Comment · Reviewer_EzZu · 2024-11-25
> > **Response to the authors**
> >
> > > Is your finding about MMLU vs perplexity published anywhere? Is it citable? None of our baseline methods report on MMLU. It is not good scholarly practice to make an anecdotal statement especially during a review.
> >
> > I apologize for omitting a reference paper in my previous comment. I understand that this may have caused some confusion. Therefore, I would like to introduce the relevant paper to support my argument.
> >
> > * Compressing LLMs: The Truth is Rarely Pure and Never Simple, ICLR 2024, https://openreview.net/forum?id=B9klVS7Ddk

---

> > > ### Author Response · Authors · 2024-11-25
> > >
> > > *"Compressing LLMs: The Truth is Rarely Pure and Never Simple, ICLR 2024, https://openreview.net/forum?id=B9klVS7Ddk*"
> > >
> > > Thank you, we now reference this paper in the discussion section (ln 529) where we state that both perplexity and downstream accuracy measures (Common Sense QA in our case) are equally important for measuring accuracy.

---

> ### Author Response · Authors · 2024-11-25
>
> *"Could you please provide the Common Sense Reasoning (CSR) or MMLU results for the comparison between the proposed method and other state-of-the-art baselines such as QuIP?"*
>
> QuIP CSR results are included in Table 5. Note that the Llama-x code by QuIP authors is broken (see e.g. https://github.com/Cornell-RelaxML/QuIP/issues/15), and we can only include the Llama-2 70B numbers for WikiText2, C4, Arc E, PIQA reported by QuIP authors.

---

> ### Comment · Reviewer_EzZu · 2024-11-25
> **Response to the authors**
>
> Thanks for the author's response. I have carefully reviewed the author's response, but my main concern about the comparison with state-of-the-art quantization methods remains unaddressed. Therefore, I will keep my score.

---

> ### Author Response · Authors · 2024-11-25
>
> *"... main concern about the comparison with state-of-the-art quantization methods remains unaddressed. "*
>
> You proposed two methods for us to compare against: FlexRound [1] and QuIP [2]. FlexRound's GitHub URL in the paper (https://github.com/onliwad101/FlexRound_LRQ) is non-functional.  As an alternative, we compared with QuIP as instructed, though only partial results are available (public QuIP code is broken).
>
> Note that AWQ and OWQ are also considered state-of-the-art, and their results on QA are in the manuscript.

---

### Author Response · Authors · 2024-11-25
**Rebuttal summary**

The reviewers raised two main concerns: 1) our proposed quantization method would not accelerate matrix-vector multiply on GPUs (Pd64), and 2) the accuracy of our quantized models should also be compared with e.g. QuIP on question-answering tasks (EzZu) along with extra ablation (z2t1). We believe we addressed both of these in a clear and reasonable manner.

For 1), we showed that our custom CUDA kernel leads to a 3.3–3.8x acceleration for the weight matrices of e.g. the OPT-175B model quantized to 3 bits per weight on average, relative to using FP16 with cuBLAS. This speed-up was measured on an Nvidia A6000 (ln 535). Our non-model-specific CUDA kernel code (~60 lines) and a short commentary are provided in Appendix A. At the last minute, Pd64 also requested a justification for our code design as well as expected performance across different hardware "options". We believe that this second request falls outside the scope of the current paper, which proposes a mathematical framework for model weight quantization.

For 2), we compared all quantization methods (proposed + 7 baselines) on three Llama-2 models against five more downstream tasks (Arc-E, Arc-C, HellaSwag, PIQA and Winogrande), showing that our method produces the highest quantized model accuracy across the majority of these tasks (table 5). Additional ablation study is included in table 2 (d). Some QuIP results are unavailable due to issues with the official QuIP code. EzZu is understandably not satisfied with this partial result, but availability of other people's code is beyond the authors' control.

---

### Meta-Review · Area_Chair_J355 · 2024-12-20

**Metareview:**

This papper introduces a convex optimization-based framework (CVXQ) for mixed-precision quantization in large language models, promising more efficient compression and competitive performance. While the theoretical approach seems solid and results on perplexity and some downstream tasks appear encouraging, all reviewers raised similar concerns.

Key issues include a lack of thorough hardware implementation details and uncertainty about real-world practicality. Comparisons against other state-of-the-art mixed-precision methods are insufficient, leaving it unclear if CVXQ truly outperforms strong existing baselines. Also, the evaluation focuses too narrowly on perplexity and would benefit from more diverse benchmarks.

Despite the authors’ efforts in their rebuttal, these concerns remain partially unadressed. The method’s potential is acknowledged, but without more robust comparisons and concrete hardware insights, it is hard to judge its true impact. Unfortunetly, the reviewers collectively feel the paper is not ready for acceptance at this time.

**Additional Comments On Reviewer Discussion:**

While authors adressed hardware implementation concerns and added ablation studies, reviewers remained skeptical about three core issues: incomplete comparisons with state-of-the-art methods (particularly QuIP), insufficient hardware implementation details across different architectures, and limited evaluation metric coverage. The discussion suggests authors made earnest efforts to address concerns through new experiments and clarifications, but couldn't fully resolve the fundamental concerns about practical implementation and comprehensive benchmarking.

---

### Decision · Program_Chairs · 2025-01-22

Reject